

SciPost Phys. Lect.Notes 26 (2021)

# Extreme boundary conditions and random tilings

**Jean-Marie Stéphan**⋆

Institut Camille Jordan, Lyon 1.

⋆ stephan@math.univ-lyon1.fr

## Abstract

Standard statistical mechanical or condensed matter arguments tell us that bulk properties of a physical system do not depend too much on boundary conditions. Random tilings of large regions provide counterexamples to such intuition, as illustrated by the famous 'arctic circle theorem' for dimer coverings in two dimensions. In these notes, I discuss such examples in the context of critical phenomena, and their relation to 1+1d quantum particle models. All those turn out to share a common feature: they are inhomogeneous, in the sense that local densities now depend on position in the bulk. I explain how such problems may be understood using variational (or hydrodynamic) arguments, how to treat long range correlations, and how non trivial edge behavior can occur. While all this is done on the example of the dimer model, the results presented here have much greater generality. In that sense the dimer model serves as an opportunity to discuss broader methods and results. [These notes require only a basic knowledge of statistical mechanics.]

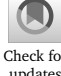

# 1 The limit shape phenomenon

Perhaps one of the greatest strength of theoretical physicists is their ability to make simplifying assumptions which are obviously not true. These untrue assumptions are then used to make powerful predictions, some of which can be confirmed by experiments or even proved mathematically. For example, band theory assumes periodic boundary conditions to get a well defined momentum, and explains why certain materials are conductors, insulators or even topological insulators. To take an even older example, the hydrodynamic description of a glass of water relies on well defined conserved quantities such as mass density or momentum density. This is obviously not true, since the boundary breaks momentum conservation. However far from the boundary this assumption is much more reasonable to the leading order, which is why nobody is (and should) be worried about this.

Similar assumptions are routinely made in classical statistical mechanics. For example, Onsager famously solved [1] the 2d Ising model, and found an exact formula for the free energy. What he did was assume periodic boundary conditions (or translational invariance), which simplify the calculations considerably. Then, his results for the free energy holds for any reasonable large chunk of the square lattice, since the free energy is well-known not to depend on boundary conditions. This simple but important observation is taught in any course

on statistical mechanics. It also motivates introducing the *dimer model*, which we will discuss at great length in the lectures. One of the most striking property of the dimer model is that it totally violates the previous well-known fact of life. The free energy of the dimer model does depend, heavily, on boundary conditions.

## 1.1 Domino tilings and boundary conditions

Let us introduce the dimer (or domino tiling) model, on the square lattice. Edges connect nearest neighbors. Dimers are entities that cover edges of the lattice. The model then asks to cover the lattice with dimers, while following the constraint that *each lattice site be touched by exactly one dimer*. We then call dimer covering a valid configuration of dimers on the lattice. For example, all 5 possible coverings of the $4 \times 2$ square lattice are shown below.

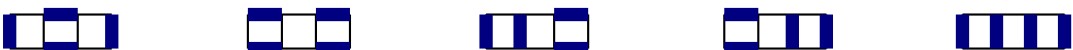

We are of course interested in the thermodynamic limit, that is, dimer coverings of large $M \times L$ lattices. As we shall see shortly the dimer model is exactly solvable; for example, the partition function has been computed by Kasteleyn [2] and Temperley and Fisher [3]. They found that the corresponding free energy is

$$F = -\frac{1}{LM} \log Z \tag{1}$$

$$\to -\frac{C}{\pi} \tag{2}$$

in the limit $L \to \infty$, $M \to \infty$, $L/M$ fixed. $C$ is the Catalan constant, $C/\pi \simeq 0.29156$. Hence the number of possible coverings grows extremely fast when $L, M$ are increased.

Examples of dimer coverings on an $L \times L$ grid are shown in figure 1. In each picture, a dimer covering is picked uniformly at random, and drawn using a color code that we explain now. The lattice is bipartite, which means one can label lattice sites with two colors, black and white, in such a way that each black (white) lattice point has all its nearest neighbors of the opposite color. Then, we draw a vertical dimer in blue (resp. yellow) if its bottom part touches a black (resp. white) vertex. Similarly green (resp. red) dimers have their left part that touches a black (resp. white) vertex. At this stage the only purpose of this convention is to make the pictures look nicer. For the same reason we also make the dimers much thicker, so that they fill all space, and the underlying lattice cannot be seen anymore.

Figure 1: Thermodynamic limit

Dimer configuration chosen uniformly at random, in the set of all possible coverings of the $L \times L$ square lattice $\{(x, y) \in \mathbb{N}^2, x < L, y < L\}$. From left to right, $L = 16, 64, 256$.

Much later, in 1991, Ref. [4] studied dimer coverings of a peculiar region

$$\mathbb{A}_L = \{x, y \in \mathbb{Z} + 1/2 \quad , \quad |x| + |y| \le L\}, \tag{3}$$

which they called *Aztec diamond*. They showed using combinatorial methods that the partition function, or number of dimer coverings, is given by the remarkably simple formula

$$Z = 2^{L(L+1)/2}. \tag{4}$$

Therefore, there are much less available dimer coverings on the Aztec diamond that there would be on a regular square grid with the same area. We will see in the following that this is an effect of boundary conditions, so free energy *does* depend on boundary conditions for dimers.

Even more remarkable is the following observation. Draw for reasonably large $L$ a dimer covering out of the $2^{L(L+1)/2}$ available ones, uniformly at random, as we did before. The corresponding pictures are shown in figure 2. The covering appears only random inside a region which looks roughly like disk for large $L$. Outside of the disk the orientations appear deterministic, with red dimers on the left, blue dimers at the bottom, etc. This is one of the

---

**Figure 2: The arctic circle theorem in action**

Dimer coverings of an Aztec diamond of order $L$, chosen uniformly at random. From left to right, $L = 16, 128, 1024$. As $L$ increases, dimers appear totally frozen outside a region, which looks like a disk. There are, however, non trivial long range fluctuations inside the disk.

---

most famous instance of what is called the limit shape phenomenon. We will give a more precise definition of limit shape later on, but for now, let us just introduce the terminology that we will use in these notes. In the deterministic region dimers are essentially frozen, so it is called *frozen region*. In contrast, dimer orientations fluctuate in the *fluctuating region*, sometimes also called liquid region. We will see later that correlations functions decay as power-laws, so the liquid region is critical. The interface between the frozen and liquid region is called an arctic curve. In the case discussed above, Ref. [5] proved that the arctic curve becomes an exact circle in the limit $L \to \infty$, a result which now goes under the name *arctic circle theorem*.

There is no a priori reason why the arctic curve should be a circle or even smooth in this particular case, this just comes out of a long calculation. Lattice symmetries, however, do impose that the arctic curve be invariant under rotations by $\pi/2$. To illustrate this last point, one can generalise the model to include "interactions" between dimers. The simplest way to do that is to put weights favouring (or disfavouring) aligned dimers on a given plaquette [6,7].

In that case the partition function may be written as

$$Z = \sum_{\mathcal{C}} e^{\lambda N_{\text{par}}}, \tag{5}$$

where $N_{\text{par}}$ counts the number of plaquettes with two dimers parallel to each other, that is, plaquettes of the form:

 ▮▮    or    ▬▬ 

$$\tag{6}$$

Positive (negative) $\lambda$ corresponds to attractive (repulsive) interactions between the dimers. For $\lambda \neq 0$ the arctic curve is not known analytically, but simulations shown in figure 3 clearly suggest that the arctic curve is very different from a circle. In fact, a closely related six vertex model has arctic curves which can be computed [8,9] and they are not even algebraic in general. See the bottom part of figure 3 for examples.

To finish this section let us mention that the pictures can be generated using a simple Markov chain Monte Carlo algorithm, which goes as follows in the free case. Start from any simple configuration, and then repeat the following lots of times:

 (i) Pick a plaquette uniformly at random.

 (ii) If it has two horizontal (resp. vertical) dimers such as shown in (6), flip them to get vertical (resp. horizontal) dimers. Otherwise do nothing.

After lots of updates, the Markov chain will thermalize and show typical configurations, which were shown above. Of course, the number of necessary updates increases quickly with system size, so generating the best pictures might take a while for large $L$. There are more complicated (but more efficient) algorithms, see e. g. [10,11] for this geometry, and [7] for more general boundary conditions.

## 1.2 Detour: a one-dimensional classical Coulomb gas model

The limit shape phenomenon may be illustrated by the following simple 1d model, which contains the basic ingredients needed to get a inhomogeneous density profile with frozen region. We consider $N$ particles living in a box $B = [-1, 1]$. The probability of having the particles at positions $x_1, \ldots, x_N \in B$ is given by

$$P(x_1, \ldots, x_N) = \frac{1}{Z_N(\beta)} \prod_{1 \le i < j \le N} |x_i - x_j|^{\beta}, \tag{7}$$

where $\beta$ is a positive real number. This probability density function may be interpreted as a Boltzmann weight $P(x_1, \ldots, x_N) = \frac{1}{Z_N(\beta)} e^{-\beta E(x_1, \ldots, x_N)}$, where $\beta$ is now inverse temperature. The energy

$$E(x_1, \ldots, x_N) = -\frac{1}{2} \sum_{1 \le i < j \le N} \log(x_i - x_j)^2 \tag{8}$$

is the electrostatic energy of a gas of $N$ particles with 2d Coulomb interactions. For this reason it is often called Coulomb gas[1]. We will consider two versions of the model:

• The first is the model as stated, with partition function

$$Z_N(\beta) = \int_{B^N} dx_1 \ldots dx_N e^{-\beta E(x_1, \ldots, x_N)}. \tag{9}$$

---

[1]By 2d we mean what would be the Coulomb potential, $V_2(\mathbf{r}_1, \mathbf{r}_2) = \log \frac{1}{|\mathbf{r}_1 - \mathbf{r}_2|}$, corresponding to solving the Poisson equation in two dimensions. Unfortunately we live in three dimension, so the real Coulomb potential is $V_3(\mathbf{r}_1, \mathbf{r}_2) = \frac{1}{|\mathbf{r}_1 - \mathbf{r}_2|}$, even when the particles are stuck to a lower dimensional region.

Figure 3: Interacting dimers and six vertex model

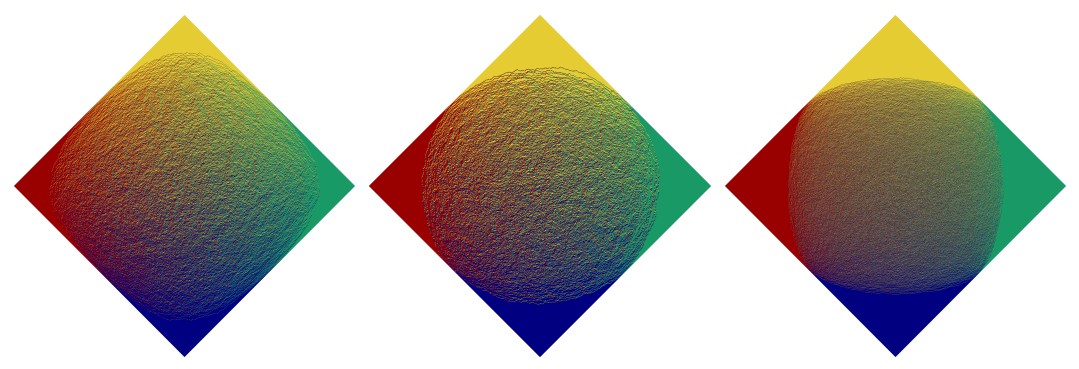

Interacting dimers on an Aztec diamond of size $L = 1024$. Left: repulsive interactions $e^\lambda = 0.4$. Middle: free dimers $e^\lambda = 1$. Right: attractive interactions $e^\lambda = 3$.

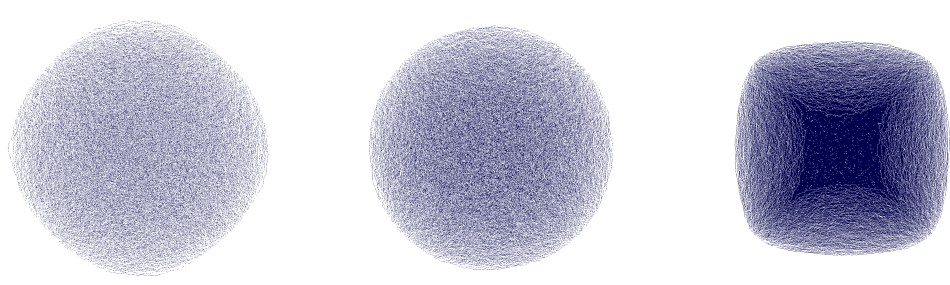

Interacting dimers on the Aztec diamond with interaction set only on plaquettes of the even sublattice. This model can be mapped to the integrable six vertex model with domain wall boundary conditions (in which case $a = b = 1$ and $\Delta = 1 - e^\lambda$) [4]. Left: $\Delta = 0.4$, middle $\Delta = 0$, right $\Delta = -6$. The arctic curve is known but complicated away from the free point $\Delta = 0$, where we are back to a circle. Here we show in blue even flippable plaquettes, or odd empty plaquettes (this corresponds to $c$ vertices in six vertex language). Note the appearance of a third region for negative $\Delta$, where most vertices are of $c$ type, and correlation functions decay exponentially. This region is usually called *gas*.

This is well known from random matrix theory, where it is called (a limit of) the Jacobi ensemble.

- In the second we impose that the allowed positions for the particles be discrete. The particles now live in $B_L = \{-1 + \frac{2j}{L-1} \; j = 0, 1, \ldots, L-1\}$. The partition function reads

$$Z_N(\beta) = \sum_{x_1 \in B_L} \cdots \sum_{x_N \in B_L} e^{-\beta E(x_1, \ldots, x_N)}. \tag{10}$$

This type of models usually go under the name discrete beta ensembles.

We are interested in the average distribution of the charges, in the limit $N \to \infty$. In the second model, we further impose a fixed density $d = N/L$. Formally, the first may be recovered from the second by considering a low density limit $d \to 0$. We study the density profile in both models separately.

Before proceeding let us point out the following important fact: the interaction is long range so energy is of order $N^2$, while entropy fluctuations are expected to be of order $N$. This is not the standard situation in thermodynamics, where there is typically a competition between energy and entropy. Here energy dominates, which means the average density profile (limit shape) can be obtained just by minimising energy. Hence, the precise value of $\beta$ does not matter here. Of course, fluctuations on top of the limit shape are still there. They are more complicated, especially in the discrete case.

### 1.2.1 The continuous gas

The continuous gas may be treated using standard techniques [12]. Introducing the density $\rho(x) = \sum_{i=1}^{N} \delta(x - x_i)$, the energy may be rewritten, up to an unimportant additive constant as

$$E(x_1, \ldots, x_N) = -\frac{1}{2} \int_{B^2} dx\, dx' \log(x - x')^2 \rho(x)\rho(x').  \tag{11}$$

[The singularity along the diagonal in the previous equation is integrable]. In the thermodynamic limit, $\rho$ is expected to become a smooth function. Therefore, finding the equilibrium distribution of the charges boils down to minimizing the energy functional

$$\mathcal{E}[\rho] = -\int_{B^2} dx\, dx' \log(x - x')^2 \rho(x)\rho(x')  \tag{12}$$

subject to the constraint

$$\int_B \rho(x)dx = N.  \tag{13}$$

Handling the constraint can be done by introducing a Lagrange multiplier $\lambda$. We consider the functional

$$\mathcal{L}[\rho, \lambda] = \mathcal{E}[\rho] + \lambda \left( 1 - \int_B dx\, \rho(x) \right),  \tag{14}$$

and write down the Euler-Lagrange (EL) equations

$$\frac{\delta \mathcal{L}}{\delta \rho(x)} = 0,  \tag{15}$$

$$\frac{\partial \mathcal{L}}{\partial \lambda} = 0.  \tag{16}$$

The second equation gives back the constraint, while the first reads

$$\int_B dx' \log(x - x')^2 \rho(x') + \lambda = 0.  \tag{17}$$

One can check that $\rho(x) = \frac{\lambda}{2\pi \log 2} \frac{1}{\sqrt{1 - x^2}}$ is a solution to the above linear integral equation. Typically, uniqueness is guaranteed by the convexity of the energy functional, which is the case here. Finally, normalization of $\rho$ yields $\lambda = 2N \log 2$. Hence we get the density profile

$$\rho(x) = \frac{N}{\pi \sqrt{1 - x^2}}.  \tag{18}$$

This density, called the arcsine law, is integrable but unbounded near $x = \pm 1$. This means the particles tend to accumulate near the two boundaries, a non trivial effect of the long-range repulsion between them. Let us finally mention that this type of problem has been studied for a very long time, in relation to potential theory as well as orthogonal polynomials. For example,

Stieltjes observed back in 1885 [13,14] that the positions of the particles that minimizes the electrostatic energy (8) on $B = [-1, 1]$ coincide with the zeroes of known orthogonal polynomials[2]. In this context, the limiting distribution (18) was probably known even before. See also exercise 1.3.

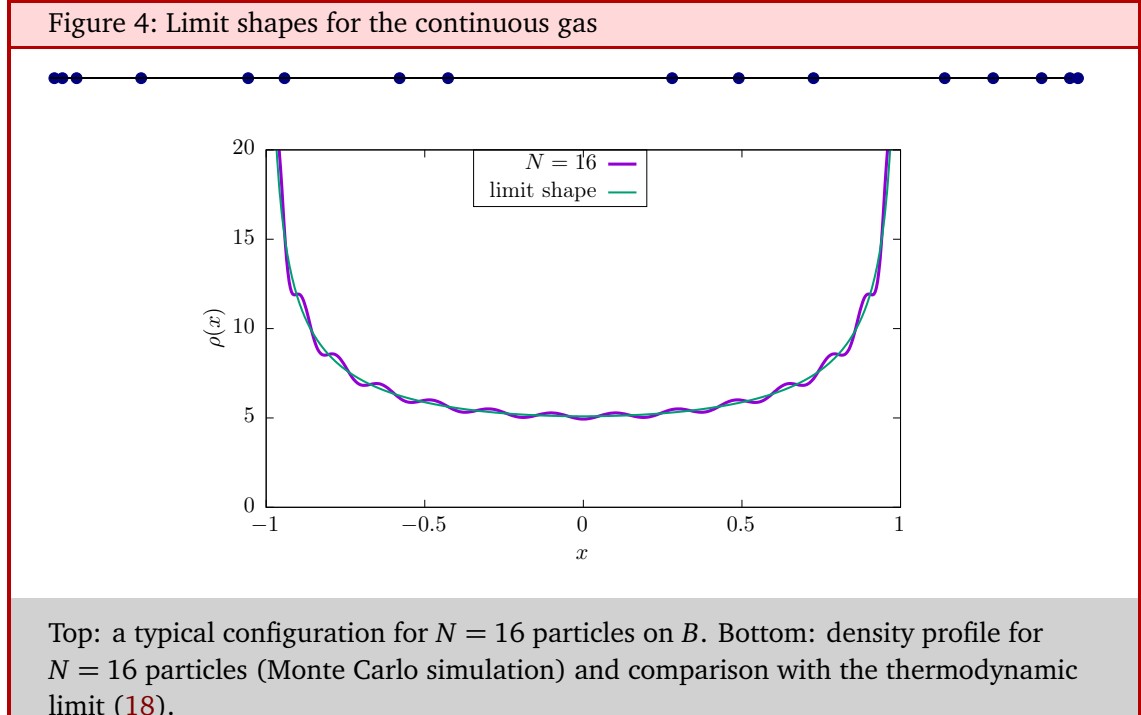

**Figure 4: Limit shapes for the continuous gas**

Top: a typical configuration for $N = 16$ particles on $B$. Bottom: density profile for $N = 16$ particles (Monte Carlo simulation) and comparison with the thermodynamic limit (18).

### 1.2.2 The discrete gas

The discrete model can be treated using a similar method. Introducing the particle density $\rho_x = \sum_i \delta_{x,x_i}$, we expect that it becomes a continuous function $\rho(x)$ normalized to $\int_B \rho(x) = 2d$ in the limit $L \to \infty, N \to \infty$, with fixed total density $N/L = d$. We still have to minimize the energy functional $\mathcal{E}[\rho]$ of (12) with the constraint (13). However, due to the discrete nature of the problem, density cannot exceed one, since two particles cannot sit on the same site. This means we get a second constraint

$$\rho(x) \leq 1 \quad , \quad \forall x \in [-1, 1]. \tag{19}$$

This extra constraint is *typical for discrete models*. The previous solution (18) is unbounded, which means a new analysis is needed. The solution to minimizing (12) with the constraints (13) and (19) was found in Ref. [17]. Let us first give the result and comment on it later. The limit shape is given by

$$\rho(x) = \begin{cases} \frac{2}{\pi} \arcsin\left(\frac{d}{\sqrt{1-x^2}}\right) & , \quad |x| < \sqrt{1-d^2} \\ 1 & , \quad \sqrt{1-d^2} < |x| < 1 \end{cases}. \tag{20}$$

---

[2]More precisely, the equilibrium positions for the pdf $\prod_{i=1}^{N}(1+x_i)^a(1-x_i)^b \prod_{j>i}(x_i-x_j)^\beta$ on $B^N$ are the $N$ roots of the polynomials $P_N^{(2\beta a-1, 2\beta b-1)}(x)$, where the $P_N^{(p,q)}$ are the Jacobi polynomials, which are orthogonal on $B$ with respect to the measure $d\mu(x) = (1+x)^p(1-x)^q dx$ $(p, q > -1)$. The interested reader may consult [15] and references therein for a broader overview. The case $\alpha = \beta = 0$ which we are dealing with here is degenerate [16], since the measure is not integrable; in this case the equilibrium positions of the particles are the zeroes of the polynomial $(1-x^2)P_{N-2}^{(1,1)}(x)$.

As in the continuum the Coulomb interaction wants to make the density very large close to the edge. This is not possible, due to the constraint (19). Hence, the system compromises by having the maximum allowed density over a larger region $[-1, -\sqrt{1-d^2}] \cup [\sqrt{1-d^2}, 1]$. In this region the position of the particles become deterministic, we call this the *frozen region* in analogy to dimers. The other region is fluctuating. See figure 5 for an illustration.

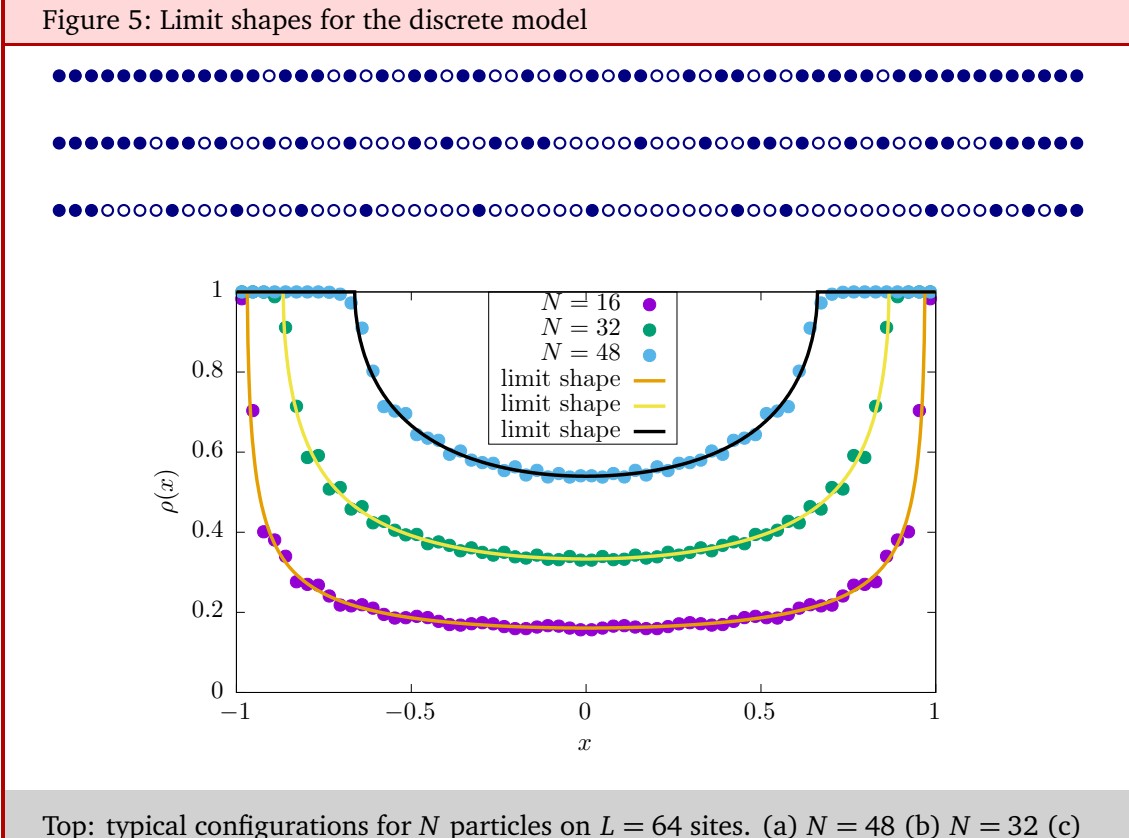

Figure 5: Limit shapes for the discrete model

Top: typical configurations for $N$ particles on $L = 64$ sites. (a) $N = 48$ (b) $N = 32$ (c) $N = 16$. Bottom (d): Corresponding density profiles and expected limit shapes (20) in the thermodynamic limit.

Let us now explain heuristically how to find the solution. The crucial point is that the second constraint (19) may be temporarily lifted by assuming that the density equals one in a given region $x \in [-1, -r] \cup [r, 1]$ close to the edges. Then, we have to minimise the new functional

$$\mathcal{E}_r[\rho] = -\int_{-r}^{r} dx \int_{-r}^{r} dx' \log(x - x')^2 \rho_r(x) \rho_r(x') + \int_{-r}^{r} \rho_r(x) V(x) dx + E_r, \qquad (21)$$

where the potential

$$V(x) = -\left( \int_{-1}^{-r} + \int_{r}^{1} \right) dx' \log(x - x')^2 \qquad (22)$$

simply comes from the interaction of the charges in $[-r, r]$ with the charges in the frozen region (in (21) there is also an additive constant $E_r$ which accounts for self-interaction in the frozen region). The constraint on the new equilibrium measure now read

$$\int_{-r}^{r} \rho_r(x) dx = 2d - 2(1 - r), \qquad (23)$$

$$\rho_r(x) \leq 1. \qquad (24)$$

Now, depending on the value of $r$, the solution of the minimization problem without imposing the second constraint may well satisfy (24) anyway (e. g for $r$ close to one it does not, while for $r = 1 - d$ it trivially does, since there is no mass left in $\rho_r$). The solution to the minimization problem (21) with only constraint (23) can be found by writing down the Euler-Lagrange equation once again. Denoting by $\rho_r$ the solution, we choose the value

$$r_0 = \sup\{r \in [0,1], \, \rho_r(x) \leq 1 \quad \forall x \in [-r,r]\}. \tag{25}$$

In words, this is the largest $r$ such that extra constraint is automatically satisfied. The solution to the minimization of (21) with constraint (23), (19) is then $\rho(x) = \rho_{r_0}(x)$ if $|x| < r$, $\rho(x) = 1$ otherwise. This yields (20). For a rigorous proof, we refer to [17].

To finish this section, let us finally mention that some examples of limit shape problems in 2d statistical mechanics can also be mapped to a similar discrete Coulomb gas problem in 1d, as was pointed out in [18]. In the following we will not pursue this direction, however, and aim for a more general hydrodynamic theory.

**Plan of the rest of the lectures.** It looks like a good idea to try and apply the general ideas we used to solve the discrete Coulomb gas model. However, it is not clear at this stage what we should minimize exactly to obtain the limit shapes in general. This question is addressed in section 2, where we introduce the height mapping and use that to understand which functional to minimize. Then, we actually compute this in section 3, for dimers on the hexagonal lattice. We use the transfer matrix formalism, and a mapping onto free fermions (see appendix A for a description of free fermions techniques, if need be). The choice of the hexagonal lattice is motivated by technical simplicity. The square lattice is similar, and left to the reader (see exercises 4.4 and 4.5). Then we use that to find the limit shapes in section 4. Section 5 deals with exact lattice calculations, which allows to recover the arctic curves in a few selected cases. Finally, we discuss a few related and more complicated problems in section 6, and conclude.

---

**EXERCISE 1.1     COVERINGS OF AN AZTEC DIAMOND     [4]**

Try (and perhaps fail) to show that (4) is correct.

---

**EXERCISE 1.2     FRACTIONAL QUANTUM HALL EFFECT**

Laughlin famously guessed [19] that the experimental observation of fractional plateaus by Tsui et al [20] may be understood by the model wave function
$\Psi(z_1, \ldots, z_N) = \prod_{1 \leq i < j \leq N} (z_i - z_j)^{\beta/2} e^{-\frac{1}{2} \sum |z_i|^2}$ where $\beta/2$ is a positive integer, the $z_i \in \mathbb{C}$, and they got the Nobel prize after that.

- - - - - - - - - - - - - - - - - - - - - - - - - - - - - - - - - - - - - - - - - - - - -

1. By interpreting $|\Psi(z_1, \ldots, z_N)|^2$ as a pdf for the $N$ particles, what is the limit shape in the limit $N \to \infty$?

2. The $N$ particles are now constrained to live on the sites of a $L \times L$ square lattice with mesh $a = 1$ (the origin $z = 0$ is set at the barycenter). What is the limit shape in the limit $N \to \infty$ with fixed density $d = N/L^2$ when $d$ is not too large? What happens at higher densities?

---

EXERCISE 1.3    ZEROS OF ORTHOGONAL POLYNOMIALS [STIELTJES 1885]

Consider the electrostatic energy considered before, with two extra charges at positions $\pm 1$, $E(x_1, \ldots, x_N) = -\sum_{i<j} \log|x_i - x_j| - a\sum_i \log|1 + x_i| - b\sum_i \log|1 - x_i|$ on $B$ for $a, b \geq 0$. For large $N$, the limit shape does not depend on $a, b$.

1. Why? In the following, we set $a = b = 1/4$.
2. Write down a system of equations satisfied by the positions $y_1, \ldots, y_N$ that minimise the electrostatic energy.
3. Let $p_N(x) = \prod_{i=1}^{N}(x - y_i)$. Show that the previous system is equivalent to $2\frac{p_N''(y_i)}{p_N'(y_i)} + \frac{1}{1-y_i} - \frac{1}{1+y_i} = 0 \quad \forall i = 1, \ldots, N$.
4. Show that $(1 - x^2)p_N''(x) - xp_N'(x) = -N^2 p_N(x)$. Show that $p_N(\cos\theta) = \cos(N\theta)$.
5. Find the zeroes of $p_N$ and recover the limit shape (18) in the limit $N \to \infty$.

---

## 2 Variational principle

We show in this section that the right quantity to minimise is a variant of the free energy. To understand this, we first introduce an important ingredient, the height mapping.

### 2.1 The height mapping

Consider the dimer model on the square lattice. Remember the lattice is bipartite, which means one can label sites with two colors, black and white, in such a way that each black (white) lattice point has all its nearest neighbors of the opposite color. To each dimer configuration we associate a height configuration, as follows. Heights are discrete numbers (integers in some units, see figure 6) which live on plaquettes (or the dual lattice, which is also square). We pick a reference point, say the leftmost bottom plaquette, and set its height to zero. Then, turning counterclockwise around a black (resp. white) vertex, the height picks up $+3$ (resp. $-3$) when crossing a dimer, $-1$ (resp. $+1$) otherwise. A dimer configuration with the corresponding height configuration is shown in figure 6 on the left. Recall the colour code used in these notes. We draw a vertical dimer in blue (resp. yellow) if its bottom part touches a black (resp. white) vertex. Similarly green (resp. red) dimers have their left part that touches a black (resp. white) vertex. Hence crossing a blue dimer from left to right, the height always picks $-3$, etc.

At the very end and for later convenience, all heights are remultiplied by $\pi/4$: in the remainder of these notes, heights are therefore elements of $\frac{\pi}{4}\mathbb{Z}$. The mapping is one to one, and has many nice properties that we will investigate in the following. For the moment let us just say that mapping to discrete heights has a long history in statistical mechanics, which dates back to Ref. [21, 22]. In fact, the model studied in these references can be mapped onto dimers on the hexagonal lattice, which we will study later on.

### 2.2 Minimizing the free energy

From the height mapping, we know that dimers like to be in configurations where the height gradient is close to zero. A good example is provided by a rectangular domain, as illustrated in figure 6 on the left. Boundary conditions might spoil that, however. The reader can easily check that the Aztec diamond geometry of figure 2 can be cooked up by imposing the following boundary condition

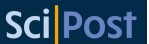

**Figure 6: The height mapping for dimers on the square lattice**

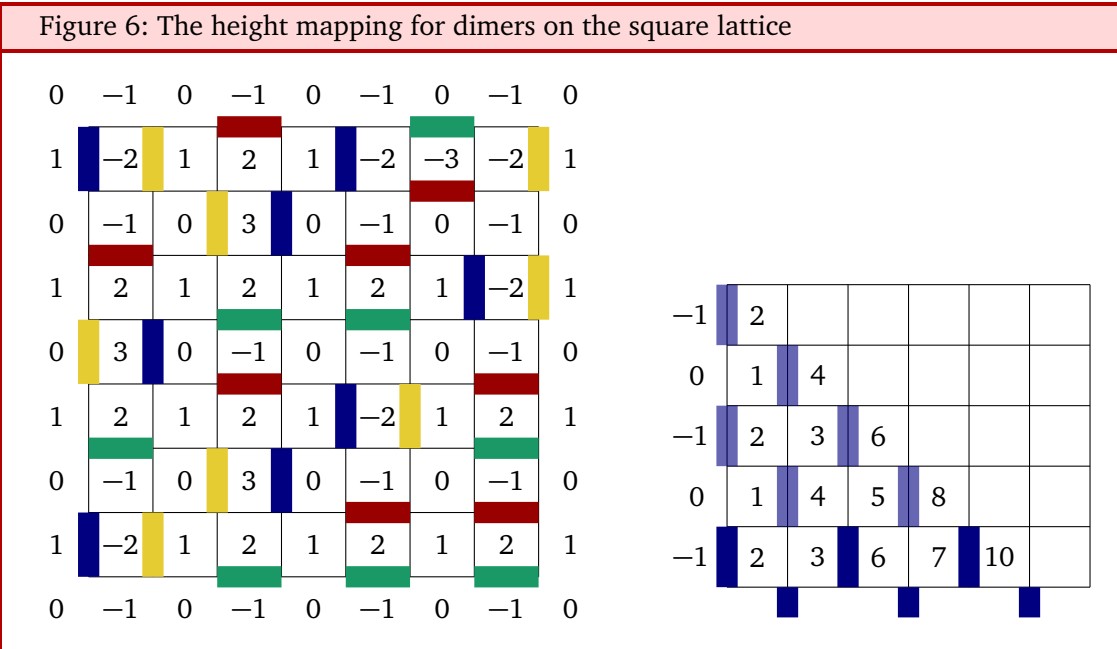

Height mapping on the square lattice, in units of $\pi/4$. Left: heights corresponding to a dimer configuration picked uniformly at random. Observe the heights remain on average quite close to zero. Right: heights corresponding to an alternating boundary condition with zigzag blue dimers. Some other vertical dimer occupancies are automatically set as a result, those are shown in lighter blue. The corresponding mean horizontal slope is maximal, $\partial_x h = \pi/2$. The gradient in the vertical direction is also automatically set to $\partial_y h = 0$. In fact, in general the slopes satisfy $|\partial_x h| + |\partial_y h| < \pi/2$.

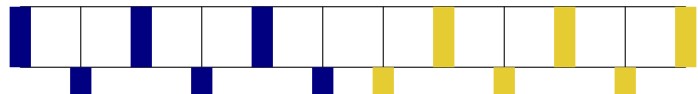

at the bottom, and a similar one at the top. The "half-dimers" in the above picture mean that the edge above is not occupied by a dimer. In that case the slopes are maximal along the Aztec diamond, and alternate $-\pi/2, \pi/2, -\pi/2, \pi/2$ from one boundary to the other.

These *extreme* boundary conditions play an important role, and are, as we shall see, responsible for the appearance of the arctic circle. Indeed, the number of available dimer coverings in a given region strongly depends on the average slopes imposed at the boundary, as can already be guessed by looking at the right part of figure 6.

A way to see this is to use the fact that all possible slopes are allowed when covering a torus with dimers, as can be seen in figure 7. So consider a $\ell \times \ell$ torus ($\ell$ even), and associate a weight $a = e^\mu$ for yellow dimers, $1/a$ for blue dimers, $b = e^\nu$ for green dimers, $1/b$ for red dimers. The corresponding partition function may be written as

$$Z(\mu, \nu) = \sum_r \sum_s Z_{r,s} \, e^{\frac{\ell^2}{\pi}(r\mu + s\nu)}. \tag{26}$$

Above, $Z_{r,s}$ precisely counts the number of dimer coverings in the $(r, s)$ sector, where $r$ is the mean horizontal slope $\langle \partial_x h \rangle$ and $s$ the mean vertical slope $\langle \partial_y h \rangle$. The total number of dimer coverings is $Z(0, 0)$. Hence $Z(\mu, \nu)$ encodes all the information about the $Z_{r,s}$. For example, the generating function reads for a $8 \times 8$ torus (311853312 coverings in total, we will explain



later how to calculate this):

$$Z(\mu, v) = 153722916 + 33490432\left(a + \frac{1}{a} + b + \frac{1}{b}\right) + 5427224\left(ab + \frac{a}{b} + \frac{1}{ab} + \frac{b}{a}\right)$$

$$+ 550928\left(a^2 + \frac{1}{a^2} + b^2 + \frac{1}{b^2}\right) + 31232\left(a^2 b + \frac{a^2}{b} + \frac{b}{a^2} + \frac{1}{a^2 b} + ab^2 + \frac{a}{b^2} + \frac{1}{ab^2} + \frac{b^2}{a}\right)$$

$$+ 1536\left(a^3 + \frac{1}{a^3} + b^3 + \frac{1}{b^3}\right) + 6\left(a^2 b^2 + \frac{a^2}{b^2} + \frac{b^2}{a^2} + \frac{1}{a^2 b^2}\right) + 4\left(a^3 b + \frac{a^3}{b} + \frac{b}{a^3} + \frac{1}{a^3 b}\right)$$

$$+ ab^3 + \frac{a}{b^3} + \frac{1}{ab^3} + \frac{b^3}{a}\right) + \frac{1}{a^4} + a^4 + \frac{1}{b^4} + b^4. \tag{27}$$

The corresponding free energy is, in the thermodynamic limit,

Figure 7: A possible height gradient on the torus

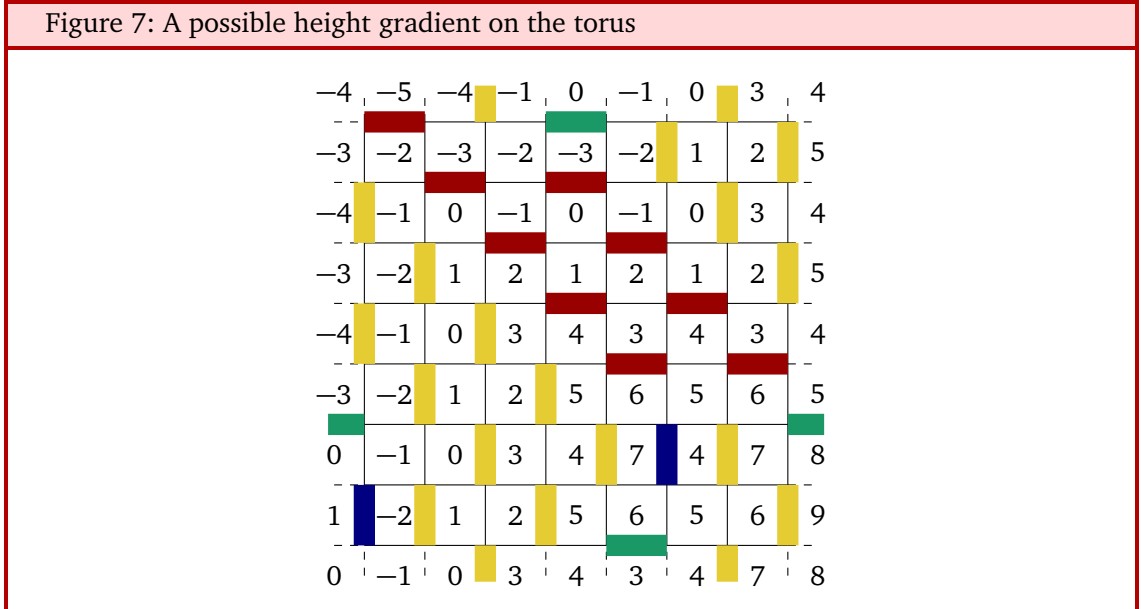

A dimer covering of the $8 \times 8$ torus, and its height configuration (in units of $\pi/4$). The fact that dimers can wrap around the torus means both $r$ and $s$ can be non zero. The slopes in this example are $r = (8/8)\pi/4 = \pi/4$, $s = (-4/8)\pi/4 = -\pi/8$. The dimer configuration drawn here contributes to the $a^2/b$ term in (27). Note that we refrain from identifying the bottom/top and leftmost/rightmost plaquettes here, for the sake of the argument.

$$F(r, s) = -\lim_{L \to \infty} \frac{1}{\ell^2} \log Z_{r,s}. \tag{28}$$

The crucial point is that once $r$ and $s$ are fixed, the resulting free energy does not depend anymore on the details of the boundary conditions. So we are back to the situation discussed in the introduction for the Ising model, provided $r$ and $s$ are fixed. This free energy (or 'surface tension') has many fascinating properties, and will play a central role in the following. For now let us just mention that it is minimum at $(r, s) = (0, 0)$; in most cases it is also a convex function.

With the free energy $F(r, s)$ as a given, one way to understand the limit shape phenomenon is to consider a very large lattice, say $L \times L$, and cut it in several much smaller (say square) cells of size $\ell \times \ell$, where $\ell$ is still much larger than the lattice spacing ($= 1$ for us). Namely, our system is macroscopic, and we look at it at mesoscopic scales where (i) it can be described

in the continuum (ii) it is uniform. The total system is then considered to be a collection of smoothly connected uniform cells, each with its own average boundary gradient $\vec{\nabla}h$, which is imposed by the surrounding cells.

The system will then try to minimize total free energy (maximize number of dimer coverings), by choosing the appropriate slopes in each mesoscopic cell. Hence we need to minimise the functional (or action)

$$S_0[h] = \int_D dx\, dy\, F(\partial_x h, \partial_y h) \tag{29}$$

over some domain $D$. The Euler-Lagrange equations for this variational problem read

$$\frac{\partial}{\partial x} F^{(10)}(\partial_x h, \partial_y h) + \frac{\partial}{\partial y} F^{(01)}(\partial_x h, \partial_y h) = 0, \tag{30}$$

where we have introduced the notation

$$F^{(ij)}(r,s) = \frac{\partial^i}{\partial r^i} \frac{\partial^j}{\partial s^j} F(r,s), \tag{31}$$

for the partial derivatives of $F$, and $h = h(x, y)$. It also possible to add extra constraints in a similar fashion to what we did in section 1.2. 

On physical grounds this variational (or hydrodynamic) principle is expected to hold under very general conditions, and has been used for a long time in the context of crystal surfaces in statistical mechanics (see e.g. [22, 23]). For dimers the situation is even more favorable, since the validity of the variational principle is now a theorem [24].

Therefore, the limit shape is given by the solution to the PDE (30) with appropriate boundary conditions. Computing exactly the free energy can also be done [22, 24], we will explain in the following section how to. It should be stressed, however, that even with an exact expression for the free energy, solving the PDE for arbitrary conditions is in general a difficult problem.

## 2.3 Fluctuations and free field theory

We have so far only discussed the limit shape, which gives the average height field at position $x$. There are of course fluctuations on top of that, which are, we argue here, described by a massless free field theory. Those fluctuations are in particular responsible for the power-law decay of correlation functions.

**Homogeneous case.** To discuss fluctuations, let us first consider the simpler case of the rectangular geometry (or really, any simply connected finite planar domain), for which $\nabla h$ is close to zero on average (see figure 6), and the slopes in the neighborhood of $(r,s) = (0,0)$ dominate. Due to lattice symmetry considerations $F(\pm r, \pm s) = F(r,s)$ and $F(r,s) = F(s,r)$, so $F^{(10)}$ and $F^{(01)}$ both vanish, and to lowest non trivial order we have

$$F(r,s) = F(0,0) + \frac{r^2 + s^2}{2\pi K} + o(r^2, s^2). \tag{32}$$

So, neglecting higher order corrections, the free energy is determined up to a single unspecified parameter $K > 0$. $K$ has many names[3], in the following we call it the *Luttinger parameter*. The

---

[3]The inverse of $K$ may be interpreted as a stiffness, for this reason $\kappa = 1/(2K)$ is called the stiffness. The terminology compactification radius $R^2 = 1/K$ comes from string theory.

Euler-Lagrange equations for the average height take a simple form in that case:

$$\left(\partial_x^2 + \partial_y^2\right)h = \nabla^2 h = 0, \tag{33}$$

so $h$ is a harmonic function with appropriate boundary conditions on $\partial D$. The solution is sometimes called the classical part of the field, and noted $h_{\text{cl}}$. In fancier terms $h_{\text{cl}}(x, y)$ is the harmonic extension of $h_\partial$, its value at the boundary, to $D$. On top of that, fluctuations may be handled by writing

$$h = h_{\text{cl}} + \frac{\delta h}{2}, \tag{34}$$

where $\delta h$ satisfies Dirichlet boundary conditions on $\partial D$ (the factor $1/2$ is set to match standard conventions later on). Plugging in (32) and dropping $F(0,0)$ yields an action

$$S[h] = S_0[h_{\text{cl}}] + S_{\text{fluc}}[\delta h], \tag{35}$$

where the linear term drops out due to the Euler-Lagrange equations, combined with the fact that $\delta h$ obeys Dirichlet boundary conditions (integrate by parts). The action for the fluctuations is

$$S_{\text{fluc}}[\phi] = \frac{1}{8\pi K} \int_D dx\, dy\, (\nabla \phi)^2. \tag{36}$$

The path integral formulation reads

$$\mathcal{Z} = e^{-S[h_{\text{cl}}]} \int [\mathcal{D}\delta h] e^{-S_{\text{fluc}}[\delta h]}, \tag{37}$$

where $\delta h$ satisfies Dirichlet boundary conditions, or, equivalently,

$$\mathcal{Z} = \int [\mathcal{D}h] e^{-S[h]}, \tag{38}$$

where $h = h_{\text{cl}}$ at the boundary.

The above is the Euclidean action of a massless free (or Gaussian) scalar field in two dimensions, in the following we will simply refer to it as the *free field*[4]. It is always desirable to visualize things, see figure 8 for two realizations of the discrete height field for dimers. The reader can then try to imagine what this becomes in the continuum limit.

Before heading to the inhomogeneous case several important remarks are in order.

- The argument we just provided is quite standard [25]; in fact, the very reason field theory techniques may be applied to statistical or condensed matter systems is that this type of reasoning often just works, and provides *exact results* for critical exponent and long range correlation functions. However, a proper derivation from a concrete lattice model is very often a difficult task.

- As can be seen from the figure, height field configurations look wilder and wilder as system size is increased. In particular, its variance at any point can be shown to diverge as $\approx \log L$. The free field is, in fact, a singular object. On a simply connected bounded planar domain $D$ a possible mathematical definition is as follows. Consider (minus) the Laplacian $-\nabla^2$ on $D$ with Dirichlet boundary conditions. We write its normalized

---

[4]It has many other names: free (compact or not) boson, bosonic string, (Tomonaga-)Luttinger liquid, $c = 1$ conformal field theory. In mathematics the names massless Gaussian field, Gaussian free field or the related Gaussian multiplicative chaos can also be found.

---

Figure 8: Discrete height field

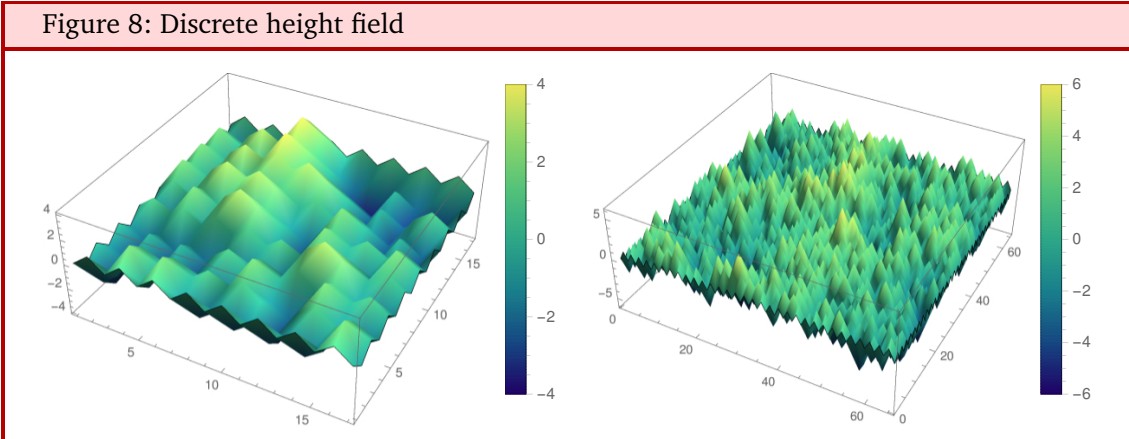

Discrete height configuration corresponding to a uniform random covering on an $L \times L$ square lattice (free boundary conditions, which means the average slopes are zero). Left: $L = 16$. Right: $L = 64$. At a given point the variance of the field can be shown to grow like $\approx \log L$.

---

eigenfunctions as $u_k(x, y)$ and the corresponding eigenvalues as $\lambda_k$ (they are all strictly positive). Then introduce

$$\varphi(x, y) = \sum_{k=1}^{\infty} \xi_k u_k(x, y), \tag{39}$$

where the $\xi_k$ are independent centered Gaussian random variables with variance $\mathbb{E}[(\xi_k)^2] = 1/\lambda_k$. This implies that $\mathbb{E}[\varphi(x, y)\varphi(x', y')] = \sum_k \frac{1}{\lambda_k} u_k(x, y) u_k(x', y')$ is the Green's function for the laplacian on $D$ with Dirichlet boundary conditions, which means $\mathbb{E}[\varphi(x, y)\varphi(x', y')] = \langle \delta h(x, y)\delta h(x', y')\rangle$, where the braket on the right denotes expectation value with respect to the path integral (37). At the level of correlations functions, $\delta h$ and $\varphi$ are the same object.

The series (39) in fact, converges almost nowhere. Therefore, the free field has to be seen as a random distribution, not a random function[5]. The interested reader can have a look at References [26–31] for a mathematical treatment of the free field. For computations what matters is to be able to integrate against smooth test functions, so this is not a problem. It is, however, nice to keep in mind that the free field is fundamentally a singular object.

- For the dimer model we will see that $K = 1$, and relate this to the fact that dimers map to free fermions. Adding local interactions (say on plaquettes, as described earlier) affects the free energy and changes the Luttinger parameter over a wide range in parameter space [6,7]. This result is proved [32,33] for sufficiently small (but finite) interaction strength. When interactions are too strong, the system typically undergoes a roughening transition [21,34] of Berezinsky-Kosterlitz-Thouless (BKT) type [35,36] to a phase where the height field becomes regular. In that case long range power law correlations are lost.

- The reader might be tempted to point out that the boundary heights are flat on average for the dimer model in a rectangular domain, so that it is reasonable to impose Dirichlet

---

[5]This follows from the Kolmogorov's three-series theorem for convergence of random series. A necessary condition for convergence of $\sum_k X_k$ in our case is that $\sum_k \text{var} X_k < \infty$, which is not the case since the eigenvalues of the laplacian decay as $k^{-2/d}$ in $d$ spatial dimensions. The analogous construction in one spatial dimension converges almost surely, and defines a brownian bridge (Brownian motion on $[0,1]$ conditioned to come back to its starting point) on $[0,1]$. So things get worse in higher dimensions.

boundary conditions $h_\partial = 0$, and safely conclude that $h_{cl}(x, y) = 0 \; \forall x, y \in D$ since this is the only harmonic function satisfying the boundary conditions. This is not quite true, in fact the correct boundary average heights alternate ($\pm \pi/8$), and this slight change has an impact on some observables in the continuum limit. $h_{cl}$ is calculated in exercise 2.2.

- The action (36) is one of the simplest example of a conformal field theory. This becomes transparent when rewriting the action in terms of complex coordinates $z = x + iy$, $\bar{z} = x - iy$, in which case $S = \frac{1}{4\pi K} \int dz d\bar{z} (\partial_z \phi)(\partial_{\bar{z}} \phi)$. One can easily check that the action is invariant under any transformation $z \mapsto g(z)$, $\bar{z} \mapsto \bar{g}(\bar{z})$, where $g$ ($\bar{g}$) is any holomorphic (antiholomorphic) function. All such transformations preserve angles locally, they are *conformal*. The action (36) is then conformaly invariant. Conformal field theory is a vast subject, we will barely scratch the surface in these notes. We refer to [37–39] for reviews.

**Inhomogeneous case.** The inhomogeneous case is slightly more complicated, due to the fact that the classical solution solves a more complicated PDE (30). The free energy is still expected to be a strictly convex function, at least over a wide range of slopes. This means the determinant of the Hessian matrix is strictly positive. This allows to still define the Luttinger parameter, as

$$\frac{1}{\pi K(r,s)} = \sqrt{\det H[r,s]}, \tag{40}$$

where $H$ is the Hessian matrix

$$H[r,s] = \begin{pmatrix} F^{(20)}(r,s) & F^{(11)}(r,s) \\ F^{(11)}(r,s) & F^{(02)}(r,s) \end{pmatrix}. \tag{41}$$

The (inverse) of the Luttinger parameter tells us how convex free energy is, so still measures stiffness. The smaller the $K$ the more energy slope fluctuations on top of the classical solution will cost. By repeating the same arguments as before, we get

$$S[h] = S_0[h_{cl}] + S_{fluc}[\delta h], \tag{42}$$

with

$$S_0[h_{cl}] = \int_D dx dy F(\partial_x h_{cl}, \partial_y h_{cl}) \tag{43}$$

and

$$S_{fluc}[\phi] = \frac{1}{8} \int dx dy (\partial_x \phi \;\; \partial_y \phi) H[\partial_x h_{cl}, \partial_y h_{cl}](\partial_x \phi \;\; \partial_y \phi)^T. \tag{44}$$

This can be written in covariant form as

$$S_{fluc}[\phi] = \frac{1}{8\pi} \int \frac{\sqrt{\det g}}{K} g^{ab}(\partial_a \phi)(\partial_b \phi), \tag{45}$$

where $g = H^{-1}$ is an emergent metric. It is a priori non flat, since $H = H[r,s]$ depends on $x, y$ through $r = \partial_x h_{cl}(x, y), s = \partial_y h_{cl}(x, y)$. What we just wrote is an action for the fluctuation $\delta h$ in a curved metric, which itself is determined from the limit shape $h_{cl}(x, y)$. A sample of the discrete height field is shown in figure 9, and illustrates what we just said.

- In the covariant form of the action (45) $K = K(\partial_x h_{cl}(x, y), \partial_y h_{cl}(x, y))$, so the Luttinger parameter depends on position in general. This has important conceptual consequences, in particular it means that conformal invariance is broken in the general inhomogeneous case. For dimers we will see in fact that $K = cst = 1$, and explain this as a general property of models that map to free fermions.

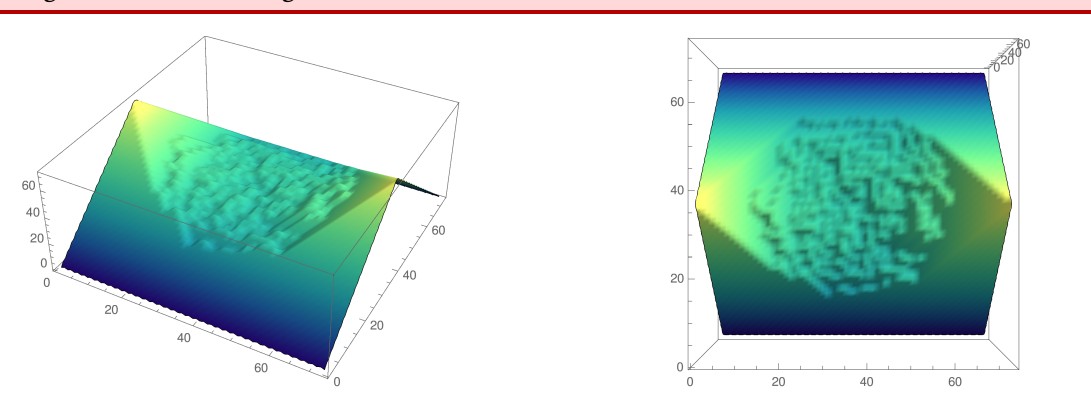

**Figure 9: Discrete height field on the Aztec diamond**

Discrete height configuration on a Aztec diamond of order $L = 72$. Left: front view. Right: top view. The average height is of order $L$, while fluctuations are of order $\approx \log L$, as before.

- The argument leading to (44) is appealing but oversimplified, for the following reason. As in the homogeneous case, we have used the EL equations to get rid of the linear term in the expansion. However, it is important to keep in mind how all those terms scale with $L$. The quadratic term is subleading (by $1/L$) compared to the linear one, and EL only implies that the leading part of the linear term vanishes, but does not tell us anything about the lower order part, which would potentially break the quadratic nature of the action. As explained in Ref. [40], a more precise argument considers the dependence of the free energy on higher derivatives $\partial^2 h$ of the height field. Then, redoing the previous steps leads to a quadratic action identical to (44), albeit with an extra ($1/L$ subleading compared to the limit shape) contribution to the average of the height field. This is not shocking if one compares to the homogeneous case, where the limit shape is zero to leading order, but the classical solution $h_{\text{cl}}$ is $O(1)$ and still matters.

- This is as far as we can go without knowing the specific form of the free energy. We compute it in the next section in the simplest example, that is dimers on the honeycomb lattice.

- Of course, obtaining exact expressions for correlations on the lattice are also very much desirable, to check our hydrodynamic assumptions and their limitations. This approach is discussed in section 5.

> **EXERCISE 2.1     SAMPLE OF THE FREE FIELD**
>
> Using your favorite programing language, draw an (approximate) sample of the free field on a rectangle $[0, \pi]^2$ with Dirichlet boundary conditions.

> **EXERCISE 2.2     THE EVEN-ODD EFFECT     [INSPIRED BY [41–45]]**
>
> We consider interacting dimers on the $L_x \times L_y$ square lattice, in the rectangular (figure 6) and cylinder geometries. $L_x$ is assumed to be even, while $L_y$ can be either even ("even case") or odd ("odd case"). We wish to compute $\mathcal{R}(\alpha) = \frac{Z(L_x, L_y) Z(L_x, L_y)}{Z(L_x, L_y+1) Z(L_x, L_y-1)}$ for

even $L_y$ in the limit $L_x, L_y \to \infty$ with fixed aspect ratio $\alpha = L_y/L_x$. To do that we assume that the only relevant contribution to the ratio of partition functions is that of the classical part $e^{-S_0[h_{\mathrm{cl}}]}$, with the action (43), meaning contributions from fluctuations are the same for even and odd.

1. Explain $\mathcal{R}(\alpha) = O(1)$ from physical arguments. Surprisingly, it turns out that $\mathcal{R}(\alpha) \neq 1$, see below.

We deal with a cylinder of circumference $L_x$ and height $L_y$ first.

2. What are the possible height differences between top and bottom contributing to $Z(L_x, L_y)$ for even $L_y$? Same question for odd $L_y$.

3. For each case, find the set of harmonic functions on the cylinder, that implement these height differences. [Hint: those are really simple functions.] Show then that

$$\mathcal{R}_{\mathrm{cyl}}(\alpha) = \frac{\sum_{n \in \mathbb{Z}} e^{-\pi n^2/K\alpha}}{\sum_{n \in \mathbb{Z}} e^{-\pi(n+1/2)^2/K\alpha}}.$$

We now treat the (more technical, use Maple/Mathematica) case of a rectangle.

5. Compute the average heights on all sides in the even and odd case.

6. Show that the real and imaginary parts of a holomorphic function are harmonic.

7. Show that the Dirichlet energy $\int_D dx\,dy\,(\nabla h_{\mathrm{e}})^2 - (\nabla h_{\mathrm{o}})^2$ is invariant under conformal maps.

The conformal map from the upper-half plane $\mathbb{H} = \{z, \mathrm{Im}\, z > 0\}$ to $D$ is given by the Schwarz-Christoffel map $w(z) = \int^z \frac{dt}{\sqrt{1-t^2}\sqrt{1-k^2 t^2}}$, $0 < k < 1$, where $k = k(\alpha)$ depends on $\alpha$. The points $z = -1/k, -1, 1, 1/k$ are mapped to the top left, bottom left, bottom right, top right vertices respectively.

8. Find the harmonic functions $\phi_{\mathrm{e,o}}(u, v)$ in the upper-half plane that implement the boundary conditions along the real axis. [Hint: with $z = u + iv$, $\arg z = \mathrm{Im}\log(z)$]

9. Show that $S[h_{\mathrm{e}}] - S[h_{\mathrm{o}}] = \frac{1}{4K}\log\frac{1-k}{1+k}$, which means $[\mathcal{R}_{\mathrm{rect}}(\alpha)] = \left(\frac{1+k(\alpha)}{1-k(\alpha)}\right)^{1/(4K)}$. Working out $k(a)$ from the map $w(z)$ leads to

$$\mathcal{R}_{\mathrm{rect}}(\alpha) = \left(\frac{\sum_{n \in \mathbb{Z}} e^{-\pi \alpha n^2}}{\sum_{n \in \mathbb{Z}} (-1)^n e^{-\pi \alpha n^2}}\right)^{1/(2K)}.$$

For free dimers ($K = 1$) the result can also be extracted from the lattice [41].

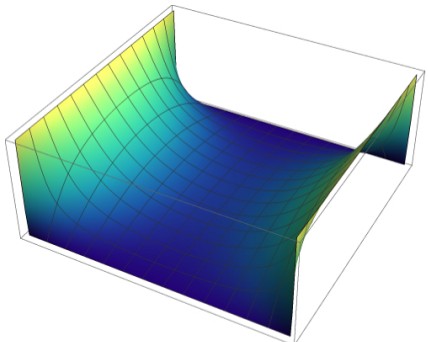
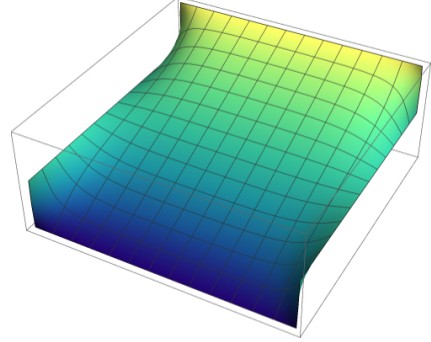

# 3 Transfer matrix for dimers

We show here how the dimer model is equivalent to a system of free fermions, through the transfer matrix formalism. Before explaining the mapping a few remarks are in order. First, the method we present is not the only one. The most widely used for dimers relies on work of Kasteleyn [2], who was the first to solve the model, expressing e.g. the partition function for dimers on planar graphs as Pfaffians (see also [3]). In particular, many results on the mathematical side rely almost exclusively on Kasteleyn theory. The two methods are not that different, see e.g. Ref [46], and lead to essentially the same results. Our choice has the advantage of connecting to techniques better known to physicists, in particular free fermions, similar to the (simplified version [47,48] of) the Onsager transfer matrix [1] of the Ising model. This choice is also motivated by possible generalizations to interacting integrable systems such as six vertex model, where the transfer matrix method cannot really be avoided. The first mapping of dimers onto free fermions is due to Lieb [49]. The version we present here is slightly different, and leads in a transparent way to a Hermitian transfer matrix with conserved number of particles, in the spirit of Refs. [7, 50, 51]. For simplicity, we mainly focus on the dimer model on the hexagonal lattice. The square lattice is similar, and worked out in exercise 4.4.

## 3.1 Reminder on the transfer matrix formalism

In this whole section we generalise the honeycomb dimer model slightly by putting alternating weights $\ldots, 1, u, 1, u, \ldots$ for horizontal dimers along a given horizontal line (see figure 10, where the lattice is drawn as a brick wall). The first important observation is that a given dimer configuration on the lattice is uniquely determined by the occupancies along vertical edges, so we may completely ignore the occupancies of horizontal ones. Put now a 0 on vertical edges

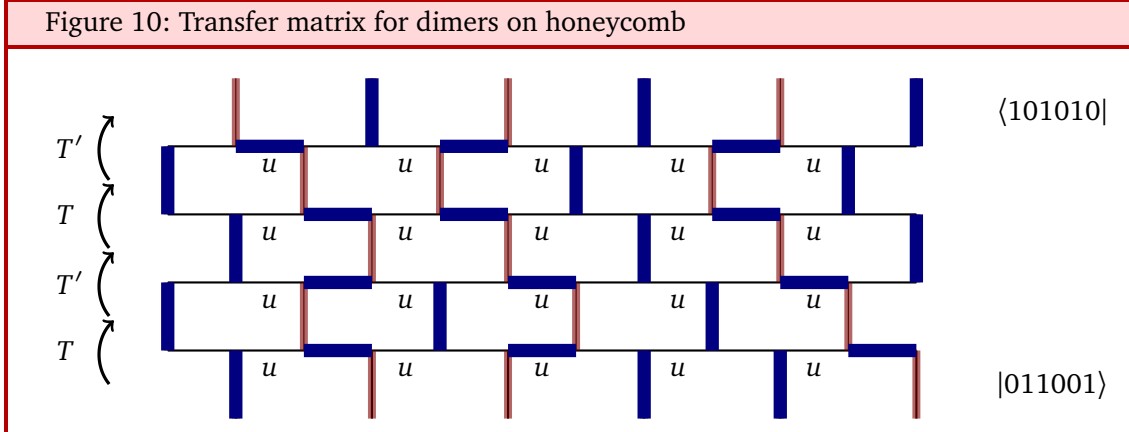

Figure 10: Transfer matrix for dimers on honeycomb

$(L = 6) \times (M = 4)$ hexagonal lattice. An example of dimer covering (dimers are thick blue lines). We see the occupancy of the top and bottom vertical edges as imposed. In thinner dark red are drawn the vertical edges not occupied by dimers, those become particles (1) in the following, while real dimers are holes (0).

occupied by a dimer (shown in thick blue on the figure), and a 1 otherwise (thinner darkred). We see the ones as a collection of particles propagating upwards, from bottom to top. We then associate a basis vector to the vertical dimers occupancies along a given horizontal line. For example for $L = 6$ in the picture, to the bottom line configuration 011001 we associate the vector $|011001\rangle$. The scalar product is $\langle \mathcal{C}|\mathcal{C}'\rangle = 1$ if the two line configurations $\mathcal{C}$ and $\mathcal{C}'$ are the same, zero otherwise. There are $2^L$ possible line configurations.

Imagine now that it is possible to find a $2^L \times 2^L$ matrix $T$, called the *transfer matrix*, such that $\langle \mathcal{C}|T|\mathcal{C}'\rangle = u^n$ if the configurations of two successive horizontal lines are compatible when stitched together –meaning they form a valid dimer covering– and zero otherwise. Here $n$ is the number of horizontal dimers with a $u$ weight when stitching the two horizontal lines. Then the partition function on the $L \times M$ lattice is simply $Z = \langle \text{top}|T^M|\text{bottom}\rangle$, where $\langle \text{top}|$ and $|\text{bottom}\rangle$ are the bras and kets corresponding to the the top and bottom configurations respectively.

*Proof.* $\langle \text{top}|T^M|\text{bottom}\rangle = \sum_{\mathcal{C}_1,\dots,\mathcal{C}_{M-1}} \langle \text{top}|T|\mathcal{C}_{M-1}\rangle \langle \mathcal{C}_{M-1}|T|\mathcal{C}_{M-2}\rangle \dots \langle \mathcal{C}_1|T|\text{bottom}\rangle$. Each element in the sum is one for valid configurations, zero otherwise, so this just counts the number of dimer coverings compatible with the top and bottom boundary condition. It is also possible to require the bottom and top boundaries to coincide (periodic boundary conditions), in which case $Z = \text{Tr}\, T^M$.

Here we actually need two transfer matrices $T$ and $T'$, since the rule changes depending on the parity of the row considered. Let us again consider the example in the figure. With a natural labeling of the edges $1,\dots,L = 6$ from left to right, the bottom configuration is $|\text{bottom}\rangle = |011001\rangle$, and $T|011001\rangle = |010101\rangle + |011001\rangle + |001101\rangle$, but we have $T'|011001\rangle = |011001\rangle + |101001\rangle + |110001\rangle + |011010\rangle + |101010\rangle + |110010\rangle \neq T|011001\rangle$.

## 3.2 Dimers as free fermions

It is important to realize that both transfer matrices (TMs) preserve the number of particles, so those are block diagonal, each block indexed by the number of particles (or ones).

**Transfer matrix in the zero particle sector.** One can easily check that $T, T'$ act as identity in this sector, namely $T'|00\dots0\rangle = |00\dots0\rangle = T|00\dots0\rangle$.

**Transfer matrix in the one particle sector.** This one is not much more difficult. The corresponding matrix elements are $A_{ij} = \delta_{i,j} + u\delta_{i,j+1}$, $A'_{ij} = \delta_{ij} + u\delta_{i+1,j}$, where

$$A_{ij} = \langle 0\dots0 \underbrace{1}_{i} 0\dots0|T|0\dots0 \underbrace{1}_{j} 0\dots0\rangle. \tag{46}$$

Here $i, j$ are the indices corresponding to the position of the (unique) one. $A'$ is similar.

**Full Transfer matrix.** For more particles the rules become more cumbersome, due to the dimer hardcore constraint, which prevents two 1 from occupying the same site. This is where the power of the free fermion formalism comes in. The reader not well acquainted with these techniques is invited to have a look at the self-contained introduction in appendix A, which has everything needed to understand the present lectures. If you already know (174), however, chances are you don't need to read it.

First, let us represent vectors $|\mathcal{C}\rangle$ using the fermion formalism. We use fermions operators to represent all states. For example for $L = 4$, we have $|1101\rangle = c_1^\dagger c_2^\dagger c_4^\dagger |0\rangle$, where $|0\rangle = |0000\rangle$ is the fermion vacuum. For any dimer configuration $\mathcal{C}$, the associated ket reads $|\mathcal{C}\rangle = c_{i_1}^\dagger \dots c_{i_n}^\dagger |0\rangle$, where the $i_1 < \dots < i_n$ are the positions of the $n$ particles (the $n$ ones) in the configuration $\mathcal{C}$. Note that since fermions anticommute, applying the operators in a different order might generate undesirable minus signs, e.g. $c_1^\dagger c_2^\dagger |0\rangle = -c_2^\dagger c_1^\dagger |0\rangle$.

The previous results read $T|0\rangle = |0\rangle$ and $Tc_i^\dagger|0\rangle = \sum_j A_{ji} c_j^\dagger|0\rangle$. Now the main claim is

(see e.g. [25])

$$T = \exp\left(\sum_{i,j=1}^{L} (\log A)_{ij} c_i^\dagger c_j\right) \tag{47}$$

provided the logarithm of $A$ makes sense. [For $T'$ just replace $A$ by $A'$ in (47)]. Hence both TMs are the exponential of a Hamiltonian that is quadratic in the fermions operator, a *free fermions Hamiltonian*.

*Sketch of the proof.* It goes in two steps. First, let us show that a sufficient condition for a good TM is that

$$T\left|\mathbf{0}\right\rangle = \left|\mathbf{0}\right\rangle, \tag{48}$$

$$T c_i^\dagger = \left(\sum_{j=1}^{L} A_{ji} c_j^\dagger\right) T. \tag{49}$$

This obviously works for $n = 0, 1$ particles. One then needs to determine the action on states $c_{i_1}^\dagger \ldots c_{i_n}^\dagger \left|\mathbf{0}\right\rangle$, $i_1 < \ldots < i_n$ in the $n$-particle sector, for $n = 2, \ldots, L$. It is obtained by commuting $T$ successively with all fermions operators using (49), and then using (48):

$$T c_{i_1}^\dagger \ldots c_{i_n}^\dagger \left|\mathbf{0}\right\rangle = \sum_{j_1, \ldots, j_n} A_{j_1 i_1} \ldots A_{j_n i_n} c_{j_1}^\dagger \ldots c_{j_n}^\dagger \left|\mathbf{0}\right\rangle. \tag{50}$$

The sum generates all possibilities for the particles to go upward-left or upward right. The jumps are not independent however, if two particles go to the same site $i$, the corresponding contribution is proportional to $c_i^\dagger c_i^\dagger = 0$, which means fermion operators effectively enforce the dimer hardcore constraint. [This observation also justifies the terminology free fermions: the particles interact only through the Pauli exclusion principle, which prevents two fermions from being in the same state (occupy the same site). In that sense free fermions are as free as fermions can ever be.] Importantly, nonzero contributions to the sum in (50) are still all ordered, since only nearest neighbor jumps are allowed in $A$. Hence all valid configurations are counted with the correct $(+)$ sign.

The second step is to realize that (47) satisfies (48), (49). This is essentially formula (171) in appendix A, go there for the proof.

**Various subtleties.**

a) The reader might be worried that the matrices $A$, $A'$ contain Jordan blocks, so are not diagonalizable. While the derivation does not require $A, A'$ to be diagonalizable, this is still a nuisance. An easy fix is to consider $T'T$ and call it the transfer matrix. Using the identity (173) we have

$$T'T = \exp\left(\sum_{i,j} (\log B)_{ij} c_i^\dagger c_j\right) \quad, \qquad B = A'A. \tag{51}$$

$B$ (and $\log B$) are now symmetric, so the Hamiltonian inside the exponential is a legitimate quantum mechanical Hermitian operator. In particular, one can use standard band theory techniques to examine its long range properties. All of them are solely determined from $T'T$, which can be considered to be the true transfer matrix. In the following, we call $T'T$ the *transfer matrix for the dimer model*.

*b)* Care must be taken when implementing periodic boundary conditions. Indeed, $Tc_L^\dagger |\mathbf{0}\rangle = (c_L^\dagger + c_1^\dagger)|\mathbf{0}\rangle$, but it is incorrect to write $Tc_2^\dagger c_L^\dagger |\mathbf{0}\rangle = (c_2^\dagger + c_3^\dagger)(c_L^\dagger + c_1^\dagger)|\mathbf{0}\rangle$, since this equals $c_2^\dagger c_L^\dagger |\mathbf{0}\rangle - c_1^\dagger c_2^\dagger |\mathbf{0}\rangle + c_3^\dagger c_L^\dagger |\mathbf{0}\rangle - c_1^\dagger c_3^\dagger |\mathbf{0}\rangle$. Indeed, remember the correspondence with the stat mech model imposes that fermion operators be applied in order. PBC spoil that natural order when the number of particles is even. Hence the transfer matrix has to satisfy, $Tc_L^\dagger = \left(c_L^\dagger + (-1)^{\hat{N}-1}c_1^\dagger\right)T$, where $\hat{N} = \sum_{j=1}^{L} c_j^\dagger c_j$ is the total fermion number operator. In practice, this means it is necessary to consider separately even and odd fermion sectors, in order to keep the quadratic nature of the Hamiltonian.

*c)* Diagonalization. As explained in appendix A.3, diagonalizing $T'T$ boils down to diagonalizing the $L \times L$ matrix $B$, which is a huge simplification. We obtain

$$T'T = \exp\left(-2\sum_{k\in\Omega}\varepsilon(k)f_k^\dagger f_k\right),\tag{52}$$

where the $\varepsilon(k)$ are the eigenvalues of $-\frac{1}{2}\log B$, $\Omega$ is the set of labels for those, and

$$f_k^\dagger = \sum_{j=1}^{L} v_{jk}c_j^\dagger,\tag{53}$$

where $v_{jk}$ are the normalized eigenvectors of $B$, $\sum_j v_{jk}v_{jq} = \delta_{kq}$. This implies

$$\{f_k, f_{k'}^\dagger\} = \delta_{kk'} \qquad , \qquad \{f_k, f_{k'}\} = \{f_k^\dagger f_{k'}^\dagger\} = 0.\tag{54}$$

For dimers on the honeycomb lattice with open boundary conditions (OBC) as described in this section, it is possible to get them explicitly. One possibility is to make the Ansatz $v_{jk} \propto \sin(kj+\gamma)$, and check that those provide unnormalized eigenvectors of $B$ provided $\gamma = 0$ and $\sin[k(L+1)]+u\sin[kL] = 0$. $\Omega$ is then the set of the $L$ solutions to the previous equation in the interval $(0, \pi)$, and

$$\varepsilon(k) = -\frac{1}{2}\log\left(1+u^2+2u\cos k\right).\tag{55}$$

The minus signs in (52),(55) are here to mimic usual band theory, where one wants to minimize energy and look for ground states, the factor 2 to remember that our transfer matrix $T'T$ moves by two steps. The case of periodic boundary conditions (PBC) turns out to be easier, and the reader can check that the set of allowed momenta is

$$\Omega_\alpha = \left\{\frac{(2m+\alpha)\pi}{L} \quad , \quad m = -L/2,\ldots,L/2-1\right\},\tag{56}$$

where $\alpha = 1$ for even $\hat{N}$ and $\alpha = 0$ for odd $\hat{N}$. This will be used in section 3.5.

## 3.3 Hamiltonian limit and quantum spin chains

We have just seen that dimers on the honeycomb lattice can be understood as a free fermion system, with dispersion (55). A similar result holds for the square lattice. For $0 < u < 1$, the dispersion relation is analytic in $[-\pi, \pi]$ so leads to a *local* Hamiltonian, that is, the hoppings decay exponentially fast with distance. Let us write $\mathcal{T}(u) = T'T$ the corresponding transfer matrix. It is easy to see that

$$\lim_{u\to 0} \mathcal{T}(u)^{1/u} = \exp(-H),\tag{57}$$

where $H$ is a free fermion Hamiltonian with dispersion $\varepsilon(k) = -\cos k$, that is a tight binding model with only next nearest neighbor hoppings. As is well known, this model also corresponds to the spin-1/2 XX chain [52]. Eq. (57) is called a Hamiltonian (or Trotter) limit, so the XX chain is a Hamiltonian limit of dimers on the honeycomb lattice (this works for the square lattice too, see figure 11).

A way to think of this limit is to consider a finite $L \times M$ lattice, and notice that $u$ essentially controls the hopping rates to the left and right. Then multiply the vertical size by an integer $p$ and divide $u$ by $p$. We have a $L \times (pM)$ lattice with hopping rate $u/p$. The limit $p \to \infty$ is nontrivial because while the hopping rate goes to zero, dimers are given more opportunities to hop since the number of application of the transfer matrix increases. It is also convenient to make the lattice spacing in the vertical direction proportional to $1/p$, so that the distance between top and bottom stays the same in that limit.

The Hamiltonian limit is in a sense intermediate between discrete and continuous. The horizontal direction stays discrete, but the vertical one becomes continuous. Hamiltonian limits are key to the application of integrability techniques to quantum spin chains [53, 54]. To finish this section let us mention the two most important examples of Hamiltonian limits: the XXZ spin chain is the Hamiltonian limit of the six-vertex model (interacting dimers), while the Ising chain in transverse field is the Hamiltonian limit of the 2d classical Ising model.

Figure 11: Hamiltonian limit of square dimers

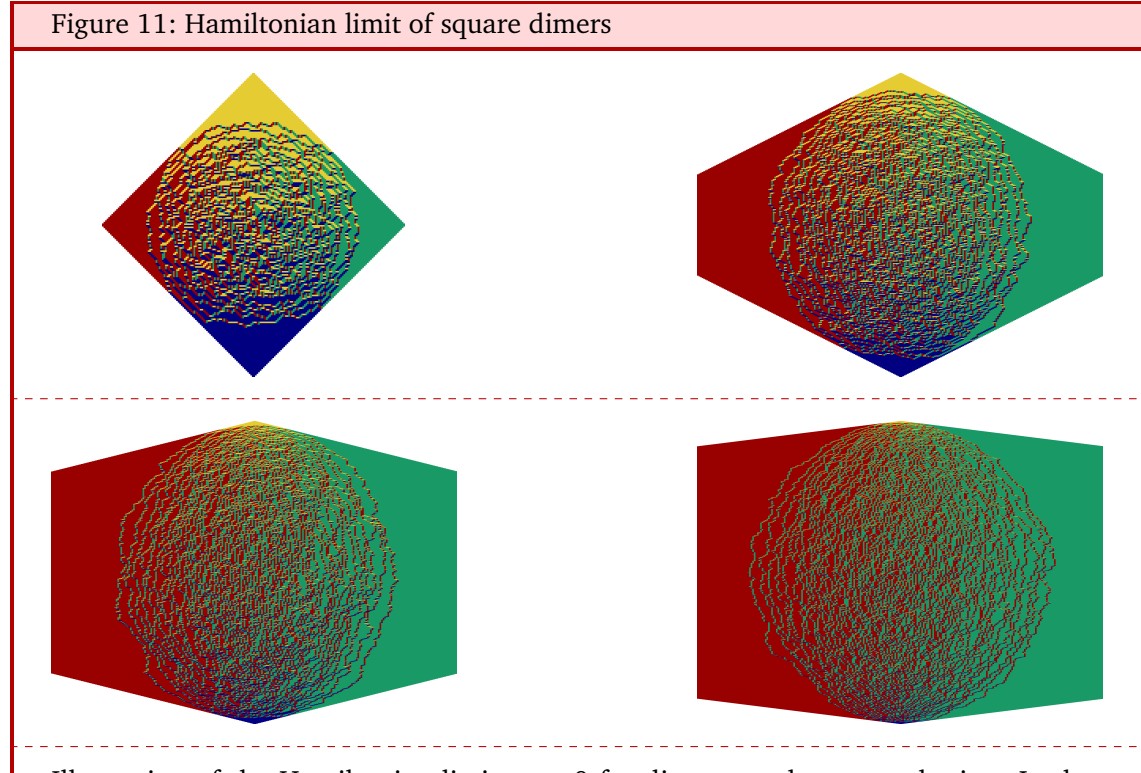

Illustration of the Hamiltonian limit $u \to 0$ for dimers on the square lattice. In that case $u$ corresponds to the weight of all horizontal dimers. Top left: $u = 1$. Top right: $u = 1/2$. Bottom left: $u = 1/4$. Bottom right $u = 1/8$. In this limit the vertical direction becomes continuous, but the arctic curve $x^2 + y^2 = R^2/(1 + u^2)$ stays a circle.

## 3.4 Connection to the height mapping

The height mapping for dimers on the honeycomb lattice is very similar to that of the square. Turning counterclockwise around black (resp. white) vertices, the height picks a factor $+2$

(resp. $-2$) when crossing a dimer, $-1$ (resp. $+1$) otherwise. As before, we use a color code to make the mapping more readable: vertical dimers always belong to the same sublattice, so we color them all in blue. Horizontal dimers are shown either in red or green. The rules are illustrated in figure 12. For future convenience, we also remultiply all heights by $\pi/3$ at the end, so that heights are integer multiples of $\pi/3$.

The relation to the free fermion mapping is as follows. First, notice that the minimum slope in the horizontal direction correspond to all edges occupied by vertical dimers, so $\partial_x h = -\frac{2\pi}{3}$. The corresponding fermion density is $n_x = 0$. The maximal slope is $\partial_x h = \frac{\pi}{3}$ and corresponds to fermion density $n_x = 1$. In the vertical direction, the minimum slope is $-\frac{\pi}{2}$ while the maximal is $\frac{\pi}{2}$. Notice again that the two slopes are not independent, $\partial_y h$ takes values in $[-k_F, k_F]$, where $k_F = \partial_x h + 2\pi/3$.

$\partial_x h$ is simply related to the fermion density, but how do we control $\partial_y h$? The simplest way to do that is to assign different weights to red and green dimers: starting from now we assign a weight $b$ to red dimers, and a weight $b^{-1}$ to green dimers. It is easy to see that the power of $b$ in the expansion of the partition function allows to determine $\partial_y h$. Then, the corresponding transfer matrix has dispersion $\varepsilon(k+iv)$, where $b = e^v$. Due to $\varepsilon(-k) = \varepsilon(k)$ the transfer matrix is still normal and real, but not symmetric anymore.

---

**Figure 12: Height mapping for dimers on the honeycomb lattice**

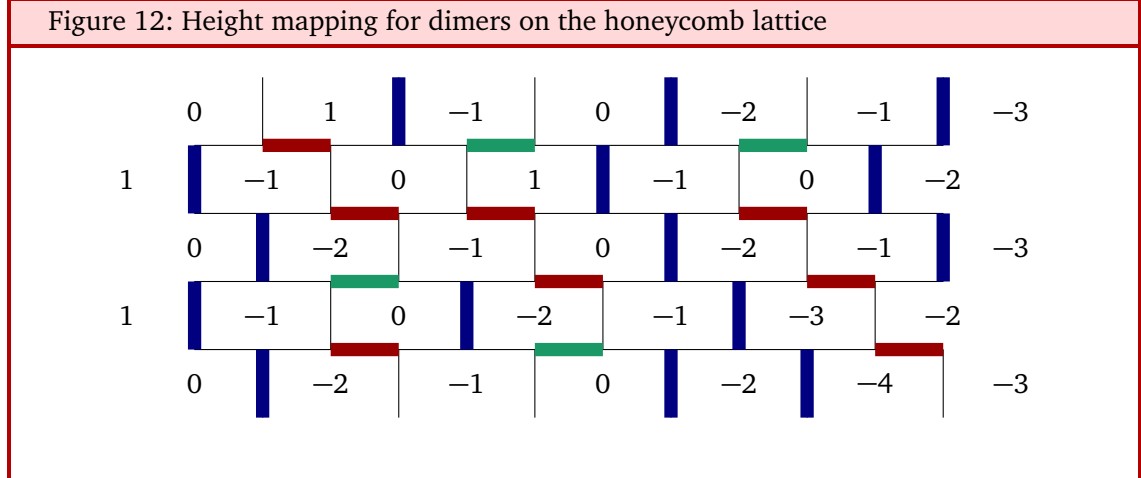

Height mapping for dimers on honeycomb, drawn as a brickwall. Heights are shown in units of $\pi/3$, as explained in the text. Recall that particles (fermions) are vertical edges not occupied by a dimer.

---

The information about heights can also be extracted in a more algebraic way, directly from the transfer matrix. The operator $c_x^\dagger c_x$ codes fermion density, and the knowledge of all on-site densities for all $x$ trivially allows to reconstruct all heights along a given horizontal line, and the average slope $\partial_x h$. This seems less obvious for $\partial_y h$, but there is nevertheless an operator which allows to do so. Assuming OBC for simplicity, it is defined as

$$J_x = \sum_{i,j=1}^{L} \Gamma_{ij}^{(x)} c_i^\dagger c_j \qquad , \qquad \Gamma_{ij}^{(x)} = \sum_{l=x}^{L} \left( \delta_{il} \delta_{lj} - B_{il} (B^{-1})_{lj} \right). \tag{58}$$

Using (171), (172), one can show that it satisfies the identity

$$\sum_{x=a}^{b} \left( T'T c_x^\dagger c_x (T'T)^{-1} - c_x^\dagger c_x \right) = J_{b+1} - J_a \tag{59}$$

for all $a, b \in \{1, \ldots, L\}$, $b \geq a$. This is similar to a continuity equation: the difference in total fermion number on the segment $[a, b]$ from one horizontal line to the next can be interpreted

as a current of particles flowing through the endpoints of the segment. $J_x$ is also local, in the sense that for any analytic dispersion, the $\Gamma_{ij}^{(x)}$ decay exponentially fast with both $|i - x|$ and $|j - x|$. Hence we call $J_x$ the *local current*.

It is also possible to make global versions of these operators. The particle density $\rho = \frac{1}{L} \sum_x c_x^\dagger c_x$ gives access to the average slope $\partial_x h$, and becomes for $L \to \infty$

$$\rho = \int_{-\pi}^{\pi} \frac{dk}{2\pi} c^\dagger(k) c(k), \tag{60}$$

with conventions explained in appendix A.5. The current density $J = \frac{1}{L} \sum_x J_x$ simplifies for large $L$ to[6]

$$J = \int_{-\pi}^{\pi} \frac{dk}{2\pi} i\varepsilon'(k + i\nu) c^\dagger(k) c(k), \tag{61}$$

and gives access to the average $\partial_y h$, as we shall check in the next subsection. Looking at (61), we immediately see that a non-zero $\nu$ is necessary to get a real non-zero current in any eigenstate of $H$.

## 3.5 Torus partition function and exact free energy for free fermions

We have now all the ingredients needed to compute the free energy. The generating function for all slopes on the $L \times M$ torus is given by

$$e^{-2ML\mu/3} Z(\mu, \nu) = \mathrm{Tr}\left[ e^{-M \sum_k [\varepsilon(k+i\nu)+\mu] f_k^\dagger f_k} \right]. \tag{62}$$

This can be evaluated in closed form, using formula (170), and being careful about the point discussed in *c)*. We find

$$e^{-2ML\mu/3} Z(\mu, \nu) = \frac{1}{2} \left( Z_0^+ - Z_0^- + Z_1^+ + Z_1^- \right), \tag{63}$$

where

$$Z_\alpha^\pm = \prod_{k \in \Omega_\alpha} (1 \pm e^{-M[\varepsilon(k+i\nu)+\mu]}), \tag{64}$$

where recall that $\Omega_\alpha$ is given by (56). The reader can check that $\Omega_1$ (resp. $\Omega_0$) allows to generate all eigenvalues in the even fermion sector (resp. odd fermion sector). The leading asymptotic behavior may be determined by picking any[7] of the four terms (64). It is then easy to see that the only terms that matter are those for which the argument of the exponential is strictly positive. Hence setting $M = L$ for convenience, we obtain

$$f(\mu, \nu) = -\lim_{L \to \infty} \frac{Z(\mu, \nu)}{L^2} \tag{65}$$

$$= -\frac{2\mu}{3} + \int_{\mathrm{FS}} \frac{dk}{2\pi} [\varepsilon(k + i\nu) + \mu]. \tag{66}$$

---

[6]For an infinite system we may analogously define the current operator as $J_x = \sum_{i,j \in \mathbb{Z}} \Gamma_{ij}^{(x)} c_i^\dagger c_j$, with $\Gamma_{ij}^{(x)} = \sum_{l=x}^{\infty} \left[ \delta_{il} \delta_{lj} - B_{il}(B^{-1})_{lj} \right]$. Now $B$ is an infinite matrix, which can easily be inverted. This leads to the integral representation $\Gamma_{ij}^{(x)} = \int_{-\pi}^{\pi} \frac{dq}{2\pi} \int_{-\pi}^{\pi} \frac{dq'}{2\pi} e^{-iqi} e^{iq'j} e^{ix(q-q')} \frac{1 - e^{-\varepsilon(q+i\nu)+\varepsilon(q'+i\nu)}}{1 - e^{i(q-q')}}$. Then, equation (61) follows from the identity $\sum_{x \in \mathbb{Z}} \Gamma_{ij}^{(x)} = \int_{-\pi}^{\pi} \frac{dq}{2\pi} e^{-iq(i-j)} i\varepsilon'(q + i\nu)$ which is obtained by applying the L'Hôpistal rule to the previous equation.

[7]Except e. g. when $u = 1$, $\nu = \mu = 0$ where $Z_0^-$ vanishes identically, since one of the $\varepsilon(k)$ is exactly zero. It can only happen to one term at a time, so does not matter.

The Fermi sea FS is the set of $k$ for which the real part of the integrand is negative, in our case it is a single symmetric interval $[-k_F, k_F]$. Therefore, the generating free energy is simply the ground state energy corresponding to the dispersion $\varepsilon(k + i\nu) + \mu$. One can check that it is real.

The last step to get the asymptotic behavior of $Z_{r,s}$ is to choose $\mu, \nu$ in such a way that the term $e^{\mu r + \nu s} Z_{r,s}$ dominates in $Z(\mu, \nu)$. In the thermodynamic limit the corresponding free energy $F(r, s)$ is given by

$$F(r, s) = f(\mu, \nu) - \frac{\mu r}{\pi} - \frac{\nu s}{\pi}, \tag{67}$$

where $\mu$ and $\nu$ are determined from

$$\partial_\mu F(r, s) = 0, \tag{68}$$

$$\partial_\nu F(r, s) = 0. \tag{69}$$

That is, $F(r, s)$ is the Legendre transform of $f(\mu, \nu)$, with $r$ (resp. $s$) conjugate to $\mu$ (resp. $\nu$). We finally obtain the following fundamental result

$$F(r, s) = \int_{-k_F}^{k_F} \frac{dk}{2\pi} \varepsilon(k + i\nu) - \frac{\nu s}{\pi}, \tag{70}$$

where $k_F$ and $\nu$ are determined from $(r, s)$ through

$$r = k_F - \frac{2\pi}{3}, \tag{71}$$

$$s = -\operatorname{Im} \varepsilon(k_F + i\nu). \tag{72}$$

Hence, the free energy is solely determined from the dispersion relation, extended to the strip $[-\pi, \pi] + i\mathbb{R}$ of the complex plane. From this formula it is also obvious that (70) is not specific to the honeycomb lattice, but applies to any one-band fermion problem, that satisfies $\varepsilon(-k) = \varepsilon(k)$. $s$ is also proportional to the mean current $\langle J \rangle$ in a Fermi sea eigenstate, since $\int_{-k_F}^{k_F} \frac{dk}{2\pi} \varepsilon'(k + i\nu) = \frac{i}{\pi} \operatorname{Im} \varepsilon(k_F + i\nu)$, further justifying the discussion in section 3.4. The result can also be generalized to several bands problems, but we do not investigate this here. Let us now compute this in two important examples.

**XX chain in imaginary time.** Let us start with the simplest, which is the PNG droplet, or XX chain in imaginary time. The dispersion relation is $\varepsilon(k) = -\cos k$, which makes explicit computations easy. We find

$$F(r, s) = \frac{s}{\pi} \operatorname{arcsinh}\left(\frac{s}{\cos r}\right) - \frac{1}{\pi} \sqrt{\cos^2 r + s^2}. \tag{73}$$

It is shown in figure 13 on the left. Now $r = k_F - \pi/2$ –notice the difference compared to dimers– so $r \in [-\pi/2, \pi/2]$, while $s \in \mathbb{R}$. The constraints between $r$, and $s$ are relaxed in this limit, except for the fact that the free energy is infinite when $r = \pm \pi/2$, $s \neq 0$.

**Dimers on honeycomb.** In this case integration is a little bit more tedious. We obtain after some algebra

$$2\pi F(r,s) = \text{Im}\left[\text{Li}_2(-ue^{ik_F-\nu}) + \text{Li}_2(-u^{-1}e^{ik_F-\nu})\right] - k_F \log u - (k_F + 2s)\nu, \tag{74}$$

where recall $\nu$ solves (72) and $\text{Li}_2(\alpha) = -\int_0^\alpha \frac{\log(1-t)}{t} dt$ is the dilogarithm. The free energy is shown in figure 13 on the right. To our knowledge, all explicitly known free energies can be expressed in terms of such dilogarithms [22, 24, 55].

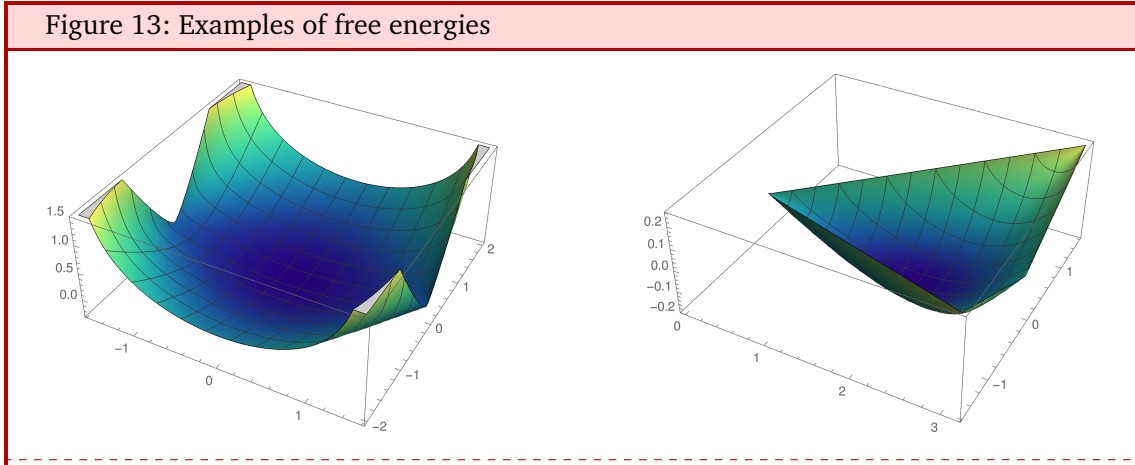

Figure 13: Examples of free energies

Left: free energy for the XX chain. Right: free energy for dimers on the hexagonal lattice, shown as a function of $k_F$, $s$, for $u = 3/5$).

## 3.6 $K = 1$ for free fermions

We are now in a position to compute the Hessian of the free energy, and check our previous claim that $K = 1$ in general for free fermions. The calculation presented below is essentially that of [56]. The first partial derivatives are

$$F^{(10)}(r,s) = \frac{\varepsilon(z) + \varepsilon(\bar{z})}{2\pi}, \tag{75}$$

$$F^{(01)}(r,s) = \frac{\text{i}(z - \bar{z})}{2\pi}, \tag{76}$$

where we have introduced $z = k_F + \text{i}\nu$. Eq. (72), or $2s = \text{i}(\varepsilon(z) - \varepsilon(\bar{z}))$ was also explicitly used to derive the second equation. Now for the second derivatives:

$$F^{(20)}(r,s) = \frac{(\partial_r z)\varepsilon'(z) + (\partial_r \bar{z})\varepsilon'(\bar{z})}{2\pi}, \tag{77}$$

$$F^{(02)}(r,s) = \frac{\text{i}\partial_s(z - \bar{z})}{2\pi}. \tag{78}$$

$F^{(11)}$ can be computed in two ways, either $\partial_r F^{(01)}(r,s)$ or $\partial_s F^{(10)}(r,s)$, so

$$F^{(11)}(r,s) = \frac{\text{i}\partial_r(z - \bar{z})}{2\pi} \tag{79}$$

$$= \frac{(\partial_s z)\varepsilon'(z) + (\partial_s \bar{z})\varepsilon'(\bar{z})}{2\pi}. \tag{80}$$

Hence

$$F^{(20)}F^{(02)} - F^{(11)}F^{(11)} = \frac{i}{4\pi^2}\big[\partial_r(z+\bar{z})\big]\big[(\partial_s z)\varepsilon'(z) - (\partial_s\bar{z})\varepsilon'(\bar{z})\big] \tag{81}$$

$$= \frac{i}{2\pi^2}\partial_s\big[\varepsilon(z) - \varepsilon(\bar{z})\big] \tag{82}$$

$$= \frac{1}{\pi^2}, \tag{83}$$

where we have again used (72) to get the last line. So $K = 1$, independent on $r$ and $s$. This identity holds for any statistical mechanical model which can be mapped onto free fermions. For more complicated models such as interacting dimers or the six vertex model, $K$ generically depends both on both $r$ and $s$ (see e. g. [33, 40, 53]).

---

EXERCISE 3.1      THE LOG GAS WITH FREE FERMIONS

Let $\psi(x_1,\ldots,x_N) = \frac{1}{\sqrt{Z_N}}\prod_{i<j}(x_i - x_j)e^{-\frac{1}{2}\sum_i V(x_i)}$ be a normalized wave function on the real line, $x_i \in \mathbb{R}$.

- - - - - - - - - - - - - - - - - - - - - - - - - - - - - - - - - - - - - - -

1. Consider the state $|\Psi\rangle = \frac{1}{\sqrt{Z_N}}f_1^\dagger\ldots f_N^\dagger|0\rangle$, where $f_m^\dagger = \int_{\mathbb{R}}dx\,x^{m-1}e^{-V(x)/2}c^\dagger(x)$ and $\{c(x), c^\dagger(x')\} = \delta(x-x')$. Show that $\psi(x_1,\ldots,x_N) = \langle 0|c(x_1)\ldots c(x_N)|\Psi\rangle$. Check that $\langle\Psi|\Psi\rangle = 1$.

2. Let $d_k^\dagger = \int dx\, p_k(x)e^{-V(x)/2}c^\dagger(x)$ where $p_k(x)$ is a polynomial in $x$ of degree at most $k - 1$. Under which condition do we have $\{d_k, d_q^\dagger\} = \delta_{kq}$ for $k, q \in \{1,\ldots,N\}$? We assume it is satisfied in the following.

3. Show that $|\Psi\rangle = d_1^\dagger\ldots d_N^\dagger|0\rangle$.

4. Show that $\langle\Psi|c^\dagger(x)c(x')|\Psi\rangle = \sum_{k=1}^{N}p_k(x)p_k(x')$.

5. Show that $\mathbb{E}(\rho(x_1)\ldots\rho(x_n)) = \det_{1\leq i,j,\leq n}(\langle\Psi|c^\dagger(x_i)c(x_j)|\Psi\rangle)$, where the expectation value $\mathbb{E}$ is taken with respect to the pdf $|\psi(x_1,\ldots,x_N)|^2$

---

# 4 Solving the minimization problem

From the previous sections, we have now all the necessary ingredients to solve the variational problem.

## 4.1 Complex Burgers equations

Recall the EL equations are

$$\partial_x F^{(10)} + \partial_y F^{(01)} = 0, \tag{84}$$

which read for free fermions, using (70,71,72),

$$\partial_x \frac{\varepsilon(k_F + i\nu) + \varepsilon(-k_F + i\nu)}{2\pi} - \partial_y \frac{\nu}{\pi} = 0, \tag{85}$$

where $k_F$ and $\nu$ depend on position $x, y$. It is possible to express $k_F, \nu$ in terms of $r, s$ but this is not necessary. A convenient way to proceed is to introduce

$$z = z(x, y) = k_F + i\nu, \tag{86}$$

as we did before. In terms of those, the EL equation reads

$$\partial_x(\varepsilon(z) + \varepsilon(\bar{z})) + i\partial_y(z - \bar{z}) = 0. \tag{87}$$

Since $r, s$ are by definition partial derivatives of the height field, we also have the continuity equations $\partial_y r - \partial_x s = 0$ which may be rewritten as $\partial_y(z + \bar{z}) - i\partial_x(\varepsilon(z) - \varepsilon(\bar{z})) = 0$. Combining with (87) yields the (conjugate of each other) equations [56]

$$i\partial_y z + \partial_x \varepsilon(z) = 0, \tag{88}$$

$$-i\partial_y \bar{z} + \partial_x \varepsilon(\bar{z}) = 0. \tag{89}$$

For $\varepsilon(k) = k^2/2$ (free Fermi gas), this PDE is called *complex Burgers equation*, in the following we refer to (88) as a complex Burgers-type equation. An alternative but equivalent point of view is discussed in section 4.5.

The interpretation of these equations is very nice. Think of a free fermion system described by a (boosted) Fermi sea $[-k_l, k_r]$, where $k_{l,r}$ are the left (or right) Fermi momenta. The particle density is $\rho = \frac{k_r + k_l}{2\pi}$, while the current is $\frac{\varepsilon(k_r) - \varepsilon(k_l)}{2\pi}$. Under unitary time evolution, they satisfy the continuity equations $\partial_t k_{l,r} + \partial_x \varepsilon(k_{l,r}) = 0$. Since $\partial_x \varepsilon(k) = (\partial_x k)\varepsilon'(k)$, this is essentially the statement that each quasiparticle with momentum $k$ moves at a speed given by the group velocity $v(k) = \varepsilon'(k)$. Now $z$ and $\bar{z}$ are the (respective) analytic continuations of $k_l$, $k_r$ to the complex plane, in which case the continuity equations are continued by Wick rotation $t = -iy$, leading to (88,89). Hence (88,89) can be seen as Wick-rotated continuity equations.

In fact Ref. [56] proceeds in exactly the reverse order as we just did. Eqs. (88,89) are just assumed to hold in imaginary time, as the only sensible analytic continuation of the real time continuity equations. Then, it is possible to find the simplest lagrangian

$$\mathcal{L} = -\frac{1}{4\pi}(z - \bar{z})(\varepsilon(z) - \varepsilon(\bar{z})) + \int_{-\bar{z}}^{z} \frac{dk}{2\pi}\varepsilon(k), \tag{90}$$

for which (88,89) are the EL equations. The reader can easily check that this is exactly the free energy (70). Here we derived all that starting from the lattice model.

## 4.2 Complex characteristics

Such equations may be "solved" by the method of complex characteristics. Without entering too much into specifics (see e.g. [57]), the solution $z$ satisfies

$$G(z) = x + i\varepsilon'(z)y, \tag{91}$$

where $G$ is an analytic function. This result is nothing more than a (nice) parametrisation of the whole set of solutions. As with all PDE's, it is very important to specify the boundary conditions; here those turn out to be enough to determine the desired analytic function $G$. Said differently, to each boundary condition (bc) there corresponds an analytic function. However, finding it from a given bc is, in general, a difficult task. At this stage the reader might be worried that we did not achieve much from a practical perspective: we mapped the difficult problem of finding the limit shape onto the difficult problem of determining the analytic function.

Fortunately the situation is not that bad, since there are a few interesting examples where $G$ can be guessed. Those include the emptiness formation probability setup [56] (see also exercise 4.1), but the method can also be adapted to a domain wall geometry, as is discussed below. Note also that in this approach, if the density profile along any horizontal or vertical slice is known, then $G$ is known, so everything is known.

### 4.3 Introducing the domain wall geometry

A simple way to generate limit shapes is to look at a 'domain wall geometry' similar to the one we used to generate the Aztec diamond for square lattice dimers. We consider lattice fermions on the infinite line $\mathbb{Z}$, and impose the configurations $|\psi\rangle = \prod_{x<0} c_x^\dagger |0\rangle$ at the bottom and top boundaries. In height language these boundary conditions correspond to the maximum slopes $\partial_x h = \frac{\pi}{3}$ for $x < 0$, $\partial_x h = -\frac{2\pi}{3}$ for $x > 0$. The boundary conditions are shown in figure 14 on the left, while a typical configuration is shown on the right.

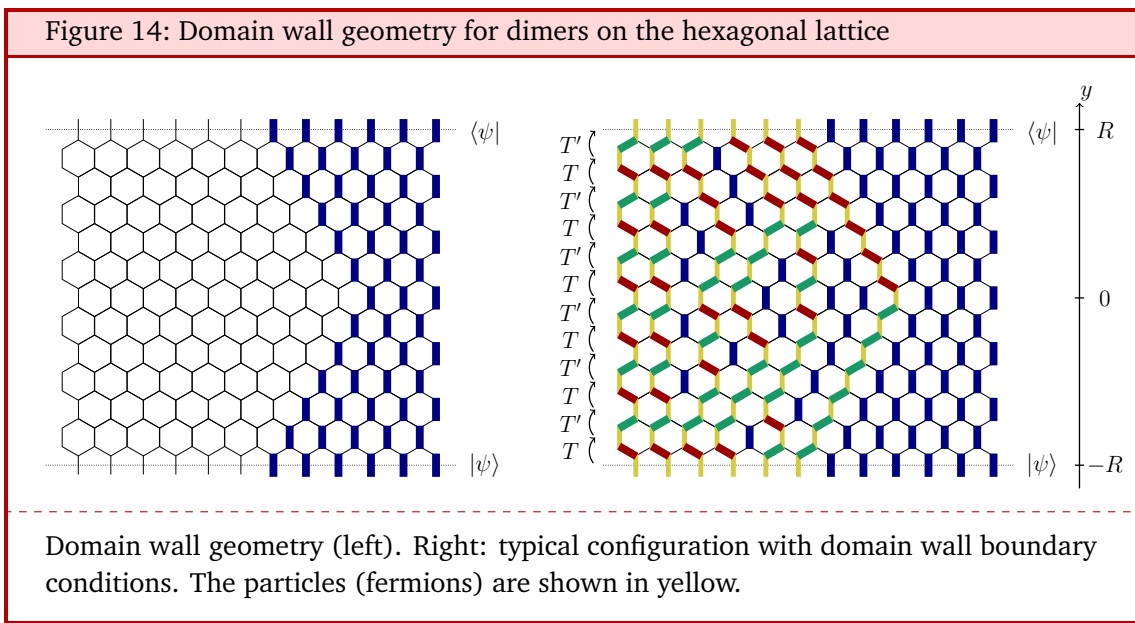

Figure 14: Domain wall geometry for dimers on the hexagonal lattice

Domain wall geometry (left). Right: typical configuration with domain wall boundary conditions. The particles (fermions) are shown in yellow.

Specific examples with this geometry are worked out in the next subsection. We will see in particular that the corresponding arctic curve is an ellipse.

### 4.4 Examples

Here we discuss two examples with boundary conditions that correspond to a domain wall initial state. We focus on the $XX$ chain first, and then proceed to dimers on the hexagonal lattice (assuming $u < 1$).

**The $XX$ chain in imaginary time (or PNG droplet [58, 59]).** Here $\varepsilon(z) = -\cos z$, which gives $\varepsilon'(z) = \sin z$. One can check the right solution for the domain wall geometry takes the form

$$R\cos z = x + \mathrm{i}y \sin z. \tag{92}$$

[This corresponds to $G(z) = R\cos z$]. It is sufficient to check that the boundary conditions are correct. Indeed, those are set at $y = \pm R$, and read (for $y = R$)

$$Re^{-\mathrm{i}k_F(x,R)+v(x,R)} = x. \tag{93}$$

The rhs is real, so this imposes $k_F(x > 0) = 0$ and $k_F(x < 0) = \pi$, which is exactly the density corresponding to the domain wall boundary conditions. (92) is a quadratic equation, so can be easily solved. Provided $x^2 + y^2 < R^2$ it has two simple roots $z$ and $-z^*$, with

$$z = \arccos\frac{x}{\sqrt{R^2 - y^2}} - \mathrm{i}\operatorname{arctanh}\frac{y}{R}. \tag{94}$$

From this we can compute essentially everything, as we demonstrate now. The density inside the arctic circle is simply given by $\rho(x,y) = \frac{\partial_x h}{\pi} + 1/2 = \frac{1}{\pi} \text{Re}\, z = \frac{1}{\pi} \arccos \frac{x}{\sqrt{R^2 - y^2}}$, the current being $-\text{Im}\, \varepsilon(z) = \frac{y \sqrt{R^2 - x^2 - y^2}}{R^2 - y^2}$. The height profile is given by

$$h(x,y) = -\sqrt{R^2 - x^2 - y^2} - |x| \arcsin \frac{|x|}{\sqrt{R^2 - y^2}}, \tag{95}$$

and the corresponding (minimal) free energy is

$$F(\partial_x h_{\text{cl}}(x,y), \partial_y h_{\text{cl}}(x,y)) = \frac{\sqrt{R^2 - x^2 - y^2}\left(-R + y \,\text{arcsinh}\left(\frac{y}{\sqrt{R^2 - y^2}}\right)\right)}{\pi(R^2 - y^2)}. \tag{96}$$

See figure 15 for pictures. The total free energy is $S_0[h_{\text{cl}}] = \int dx\, dy\, F(\partial_x h_{\text{cl}}(x,y), \partial_y h_{\text{cl}}(x,y)) = -R^2/2$, which means the partition function scales as $Z(R) \sim e^{R^2/2}$. In fact, we will see in section 5 that $Z(R) = e^{R^2/2}$ exactly for all $R$.

Figure 15: Minimisers for the XX chain

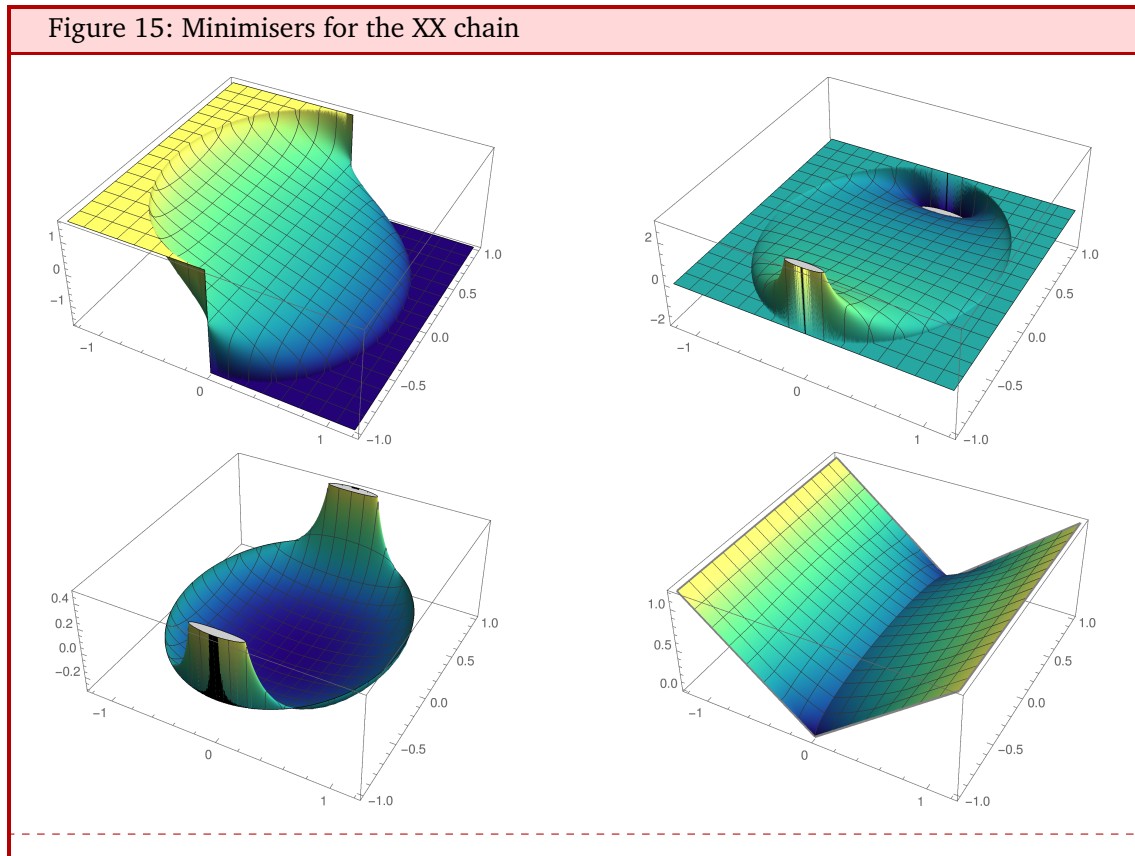

Density profile, current, free energy in the fluctuating region, (minus the) height profile corresponding to the minimiser of $\int_D dx\, dy\, F(\partial_x h, \partial_y h)$.

Looking at the edge behavior is also worthwhile. Except near $x = 0$, $y = \pm R$, the density profile vanishes with a square-root singularity. The height field in turn vanishes as $x^{3/2}$, a result which is known more generally as the Pokrovsky-Talapov law [60].

**Hilbert transform and dimers on the hexagonal lattice.** For a general dispersion relation $\varepsilon(z) = -\sum_p a_p \cos pz$, $v(z) = \varepsilon'(z)$, it is easy to see that the right generalisation of (92) reads

$$-R\tilde{v}(z) = x + i y v(z), \tag{97}$$

where $\tilde{v}$ is obtained from $v(z) = \sum_p p a_p \sin pz$, by replacing all $\sin pz$ by $-\cos pz$. This transformation is called *Hilbert transform*. Applying this to the dispersion (55) leads to

$$\tilde{v}(z) = -\frac{u(u + \cos z)}{1 + u^2 + 2u \cos z}. \tag{98}$$

We also get a quadratic equation, whose solution is

$$z = \arccos\left(\frac{(1 + u^2)x - Ru^2}{u\sqrt{(R - 2x)^2 - y^2}}\right) - i\operatorname{arctanh}\left(\frac{y}{R - 2x}\right) \tag{99}$$

inside the arctic ellipse $X^2 + y^2 < R^2$, where $X = \frac{1 - u^2}{u}x + Ru$. It is also possible to compute everything as we did before, but we refrain from doing so.

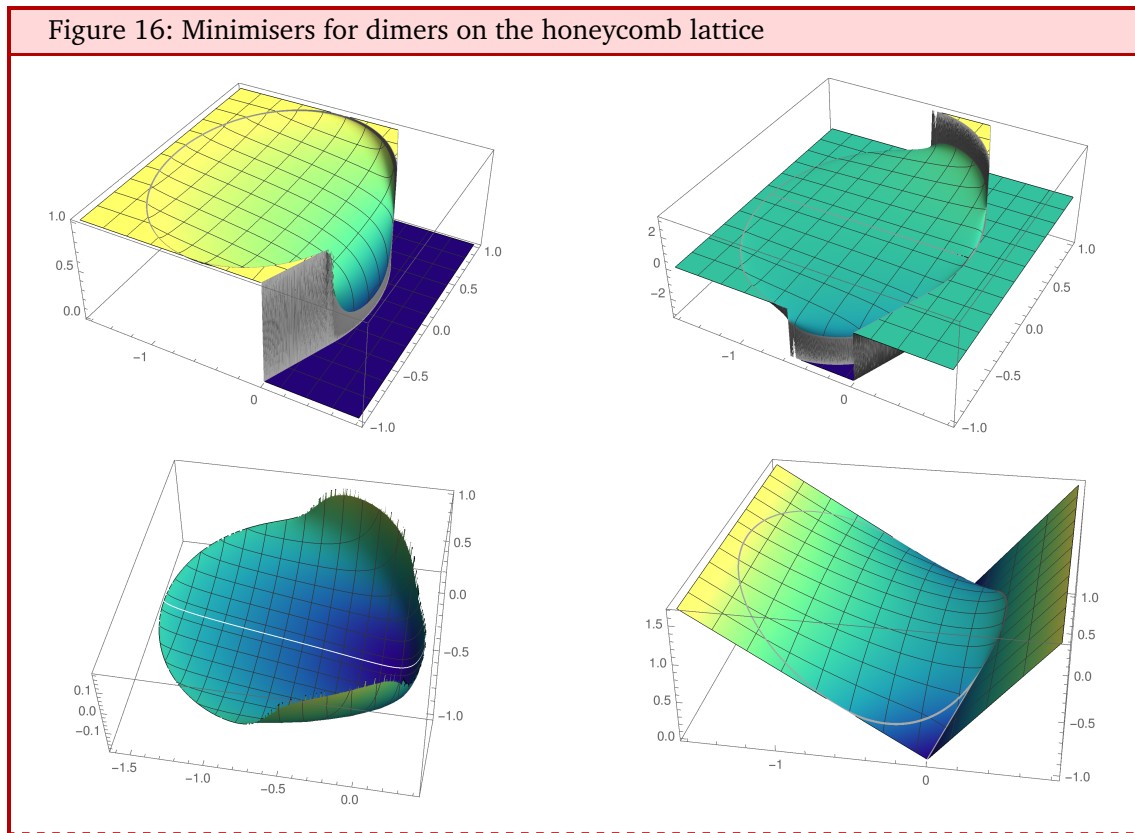

Figure 16: Minimisers for dimers on the honeycomb lattice

Density profile $\frac{1}{\pi}\partial_x h + \frac{2}{3}$, current $\partial_y h$, free energy in the fluctuating region and (minus the) corresponding height profile.

## 4.5 Algebraic geometry

Recall the free energy may be put into the form

$$F(r, s) = f(\mu, v) - \frac{\mu r}{\pi} - \frac{\mu s}{\pi}, \tag{100}$$

where

$$f(\mu, v) = \int_{-\pi}^{\pi} \frac{dk}{2\pi} [\varepsilon(k + iv) + \mu], \tag{101}$$

and $\mu, v$ are determined from (71,72), namely, as the (double) Legendre transform of the generating function $f(\mu, v)$. Let us take the honeycomb dimer model as an example. There

is an alternative (equivalent) way of accessing the generating function, using the Kasteleyn approach [22,42], which leads to the more symmetric expression

$$f(\mu, \nu) = \int_{-\pi}^{\pi} \frac{dk}{2\pi} \int_{-\pi}^{\pi} \frac{dq}{2\pi} \log \left| P(e^{\mu/\pi + ik}, e^{\nu/\pi + iq}) \right|, \tag{102}$$

where $P(z, w) = z + w + 1$. The reader can check that the two expressions match, using $\int_{-\pi}^{\pi} \frac{d\theta}{2\pi} \log(\alpha + \beta e^{i\theta}) = \log \max(|\alpha|, |\beta|)$. In a sense, the transfer matrix approach does the integral over $q$ for free in (102), but depending on context, the symmetric form can be nicer, and this is the way free energy is calculated in the mathematical literature, see e.g. [24]. The polynomial $P$ essentially encodes the lattice, more complicated ones leads to higher order polynomials, for example $P(z, w) = 1 + z + w - zw$ for the square lattice. In Kasteleyn's approach, $P$ is related to the determinant of the Kasteleyn matrix.

In algebraic geometry context, $f(\mu, \nu)$ as defined is (102) is called the *Ronkin function* of the polynomial $P(z, w) = z + w + 1$, and the free energy $F(r, s)$ is then its Legendre dual. The dual takes values inside a polygon of allowed slopes, which is the *Newton polygon* of the polynomial $P$. One can also define the *Amoeba* of the algebraic curve $P(z, w) = 0$, as the set

$$\mathcal{A}(P) = \{(\log |z|, \log |w|) \in \mathbb{R}^2, \ P(z, w) = 0\}, \tag{103}$$

namely, the projection of the algebraic curve (in $\mathbb{C}^2$) $P(z, w) = 0$ to $\mathbb{R}^2$ by the map $(z, w) \mapsto (\log |z|, \log |w|)$.

Now, from general algebraic geometry machinery, the Ronkin function is convex, and linear in the complement of the Amoeba. The complement is an union of disjoint simply connected pieces, which correspond to frozen regions [22]. Under the Legendre duality each component of the complement is mapped to a single point of the Newton polygon. One can also show that the Hessian of the Ronkin function is constant $\det \text{Hess}(R) = \pi^2$ for any point inside the Amoeba. This is interpreted as a Monge-Ampère equation. By Legendre duality this implies $\text{Hess}(F) = 1/\pi^2$, so $K = 1$ in the fluctuating region. This result happens to be a characterisation of algebraic curves known as Harnack curves, so the algebraic curves one gets from dimer models are always Harnack. The interested reader may have a look at references [57,61,62] for a much more precise discussion.

Hence, the statement "The free energy of the dimer model is the Ronkin function of a Harnack curve, so satisfies a Monge-Ampère relation for any point in the Amoeba" translates for us into the statement "Dimers map to free fermions, so bosonising the fermions we get $K = 1$ in the fluctuating region". A nice illustration of the two different types of jargons.

## 4.6 Edge behavior

The hydrodynamic solution gave $z = z(x, y) = k_F + i\nu$ from which the limit shape follows. Our aim is to provide a heuristic derivation of the universal edge behavior near the arctic curve. To do that, we look at the particular case where $\nu$ (or the current) is zero, which occurs at least in the middle at $y = 0$ in all the examples we discussed. Let us emphasize that the argument we present here may be generalized to $\nu \neq 0$. However, the discussion would involve left/right ground states of non-Hermitian Hamiltonians, which would obscure the argument slightly. We now assume that $z(x, y) = k_F(x, y)$ is known from the hydrodynamic solution and happens to be real. The first claim is that the correlations around a given point $(x, y)$ are those of the ground state of the Hamiltonian

$$H = \int_{-\pi}^{\pi} \frac{dk}{2\pi} \left[ \varepsilon(k) + \mu \right] c^{\dagger}(k) c(k), \tag{104}$$

where $\mu$ is set such that $\varepsilon(k_F) = -\mu$, to ensure that $k_F$ be the Fermi momentum.

Close to the arctic curve the density goes to 0 or 1, let us focus on the case of vanishing density. This means $k_F$ goes to zero, and it is possible to expand the dispersion around $k = 0$. Since $\varepsilon(-k) = \varepsilon(k)$, we have $\varepsilon(k) = \varepsilon(0) + \frac{1}{2}\epsilon''(0)k^2 + O(k^4)$, and we get the effective edge Hamiltonian

$$H_{\text{edge}} = \int_{\mathbb{R}} \frac{dk}{2\pi} \frac{1}{2}\varepsilon''(0)\left(k^2 - k_F^2\right) c^\dagger(k)c(k). \tag{105}$$

Now how does $k_F$ depend on $x, y$ close to the edge? The edge corresponds to the case where two (or more) solutions $z_s$ and $-z_s^*$ to the hydrodynamic equation

$$x + iy\varepsilon'(z) + R\tilde{\varepsilon}'(z) = 0 \tag{106}$$

coalesce, so that we get a double root or higher. *Generically* this root will be only double, meaning $\varepsilon''(0) \neq 0$. Writing $x_a(y, R)$ for the arctic curve, and setting $x = x_a - \tilde{x}$, we get that, generically

$$k_F(x) \sim \alpha \sqrt{\frac{\tilde{x}}{R}} \tag{107}$$

for some coefficient $\alpha$ that depends on $x_a/R$. (for example for the domain wall XX chain $\alpha = \sqrt{2}$). Therefore, $k_F^2$ behaves linearly in $\tilde{x}$, close to the edge.

Now what are free fermions with $k^2$ dispersion? This is just a free Fermi gas in the continuum, sometimes encountered via its relation to the Tonks-Girardeau gas in cold atom systems [63]. Making crudely the substitution $k \to -i\frac{d}{d\tilde{x}}$ and undoing the Fourier transform yields the Hamiltonian

$$H_{\text{edge}} \propto \int_{\mathbb{R}} d\tilde{x}\, c^\dagger(\tilde{x})\left(-\frac{d^2}{d\tilde{x}^2} + \frac{\alpha\tilde{x}}{R}\right)c(\tilde{x}), \tag{108}$$

that is free Dirac fermions $\{c(\tilde{x}), c^\dagger(\tilde{x}')\} = \delta(\tilde{x} - \tilde{x}')$, in a linear potential. The change of variables $\tilde{x} = (R/\alpha)^{1/3}u$ leads to

$$H_{\text{edge}} \propto \int_{\mathbb{R}} du\, c^\dagger(u)\left(-\frac{d^2}{du^2} + u\right)c(u). \tag{109}$$

Therefore, a non trivial Hamiltonian emerges at distances $\sim R^{1/3}$ from the edge. We expect correlations on such distance to be that of the ground state of (109). This is still a free problem, so diagonalizing $H_{\text{edge}}$ boils down to solving the single-particle eigenvalue equation $\left(-\frac{d^2}{du^2} + u\right)f(\lambda, u) = \lambda f(\lambda, u)$. The solutions are well-known to be Airy functions. It is an exercise to show that the (Dirac sea-like) ground state propagator for this Hamiltonian is

$$\langle c^\dagger(u)c(u')\rangle = \int_0^\infty d\lambda\, \text{Ai}(u + \lambda)\text{Ai}(u' + \lambda), \tag{110}$$

which is known as the *Airy kernel*.

Let us now look at a region $A = [s, \infty)$ of the infinite line. It is another nice exercise to compute the generating function $\Upsilon(\alpha) = \langle e^{\alpha \int_s^\infty c^\dagger(u)c(u)du}\rangle$, and show that it is given by the infinite series

$$\Upsilon(\alpha, s) = 1 + \sum_{n=1}^\infty \frac{(e^\alpha - 1)^n}{n!} \int_s^\infty du_1 \ldots \int_s^\infty du_n \det_{1 \leq i,j \leq n}\left(\langle c^\dagger(u_i)c(u_j)\rangle\right). \tag{111}$$

Now define the emptiness formation probability $E(s) = \lim_{\alpha \to -\infty} \Upsilon(\alpha, s)$, which is, as its name indicates, the probability that the interval $A = [s, \infty)$ be empty of particles. We have from the previous formula the exact series representation

$$E(s) = 1 + \sum_{n=1}^{\infty} \frac{(-1)^n}{n!} \int_s^{\infty} du_1 \ldots \int_s^{\infty} du_n \det_{1 \le i,j \le n} \left( \langle c^{\dagger}(u_i) c(u_j) \rangle \right). \qquad (112)$$

One can show that $E(s)$ is smooth, positive, strictly increasing, and $\lim_{s \to -\infty} E(s) = 0$, while $\lim_{s \to \infty} E(s) = 1$. $E(s)$ gives us information about the last (or rightmost) particle. Indeed $E(s + ds) - E(s)$ is proportional to the probability that the rightmost particle lies in the interval $[s, s + ds]$. Hence $p(s) = \frac{dE}{ds}$ is actually the probability density function for the rightmost particle, so $E(s)$ may be interpreted as the cumulative distribution for the rightmost particle.

The distribution that has the rhs of (112) as a cumulative distribution has a name. It is called the *Tracy-Widom (T-W) distribution* [64]. We finish with a few remarks:

- Exact series such as (112) or the one above are called Fredholm determinants. They are the continuum analog of the regular determinant, for operators. See e.g. (180,181) for a discrete analog.

- Proving that the distribution of the rightmost fermion does converge, after proper rescaling, to the T-W distribution requires of course more work than the heuristic argument we just gave. However, it still illustrates the physical mechanism through which T-W emerges, that is free fermions in a linear potential. We will see in the next section another mechanism through which T-W occurs.

- Convergence to T-W has been proved in all the models we considered so far. For the $XX$ chain this was done by Praehoffer and Spohn [59], while dimers have been treated by Johannson (honeycomb [18] and square [65]).

- The attentive reader might have spotted a physical flaw with the argument we just made. We have implicitly assumed separation of scales throughout. This is fine in the bulk, but it is not clear that it still holds at the edge. In fact, one can show that the edge is exactly borderline with respect to separation of scale, since the density varies on distances that are comparable to inter particle distances. In that sense T-W is smoothly connected to the bulk, where separation of scale does hold. This also justifies the need for exact calculations starting from the lattice, see section 5.

- The word generic near (107) is important. It is necessary for the density to vanish as square root to get a $k^2$ dispersion and a linear potential for the fermions, so $R^{1/3}$ behavior and T-W. If the arctic curve has cusps for example, edge correlations near (generic) cusps are described by a higher order kernel known as Pearcey kernel. We refer to [66] for a discussion of all these kernels. Another example of higher order kernel can be found in exercise 4.3.

- Below is a plot of the T-W probability density function. As can be observed it superficially resembles a gaussian, but it is slightly skewed.

- T-W scaling is part of a broader subject, which goes under the name Kardar-Parisi-Zhang (KPZ) equation [67] and KPZ universality class. Roughly speaking, the KPZ equation is a stochastic PDE that models interface growth [68, 69], and it turns out the long time limit of this equation with certain initial conditions is exactly the T-W distribution. We refer to [25, 70–73] for reviews on this equation, the KPZ universality class, and related topics in random matrix theory.

Figure 17: Tracy-Widom distribution

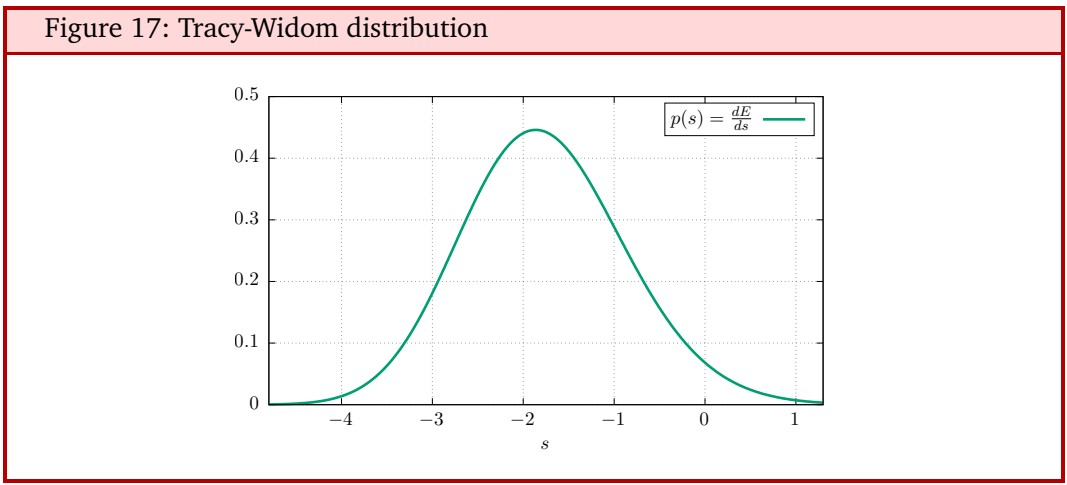

---

EXERCISE 4.1     EMPTINESS BOUNDARY CONDITIONS     [56]

We consider fermions with dispersion $\varepsilon(z) = -\cos z$. We have seen in the text, that solutions to complex Burgers may be put under the form $\cos z = G(x + i y \sin z)$, where $G$ is analytic. We wish to find the limit shape corresponding to emptiness boundary conditions, that is (i) vanishing density on an interval $x \in [-\ell, \ell]$, $y = 0$ of the full complex plane and (ii) density $1/2$ at infinity.

- - - - - - - - - - - - - - - - - - - - - - - - - - - - - - - - - - - - - - - - - - - - - - - - -

1. What is the correct function $G$ to implement those boundary conditions?
[Hint: the full density profile is shown in the picture below for $\ell = 1$]
2. Where do you expect Pokrovsky-Talapov-Tracy-Widom behavior?

- - - - - - - - - - - - - - - - - - - - - - - - - - - - - - - - - - - - - - - - - - - - - - - - -

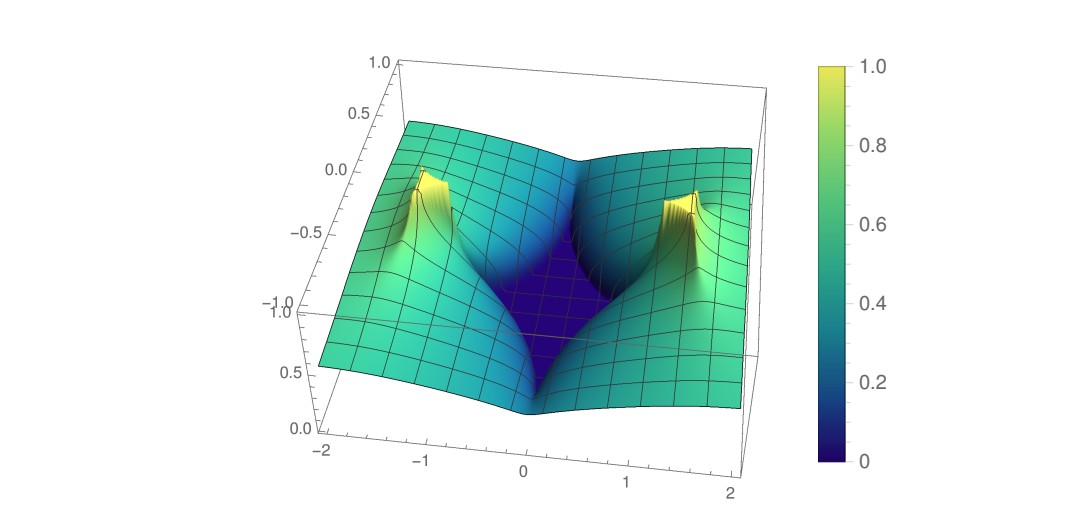

EXERCISE 4.2     REVERSE ENGINEERING OTHER SOLUTIONS

By playing with simple functions, find other solutions to the complex Burgers equation and interpret them.

EXERCISE 4.3       ANOTHER HIGHER ORDER EDGE KERNEL       [74, 75]

We consider the Hamiltonian

$$H = \int_{-\pi}^{\pi} \frac{dk}{2\pi} \big[ \varepsilon(k) - \mu \big] c^{\dagger}(k) c(k)$$

where, semiclassically, $\mu$ depends position as $\mu = \mu(x) = x/R$ for large $R$. The dispersion relation is $\varepsilon(k) = -\cos k + \frac{1}{4} \cos(2k)$.

- - - - - - - - - - - - - - - - - - - - - - - - - - - - - - - - - - - - - - - - - - - - - - - - - -

1. Where is the location of the right edge?
2. On which scales to you expect the distribution of the righmost fermion to be?
3. Compute the associated edge kernel and edge distribution.

EXERCISE 4.4       THE AZTEC ARCTIC CIRCLE (I: TRANSFER MATRIX)

The aim of this exercise is to work out the transfer matrix for dimers on the square lattice. This will be used in exercise 4.5 to recover the arctic circle shown in figure 2, and discussed at length in the introduction.

The mapping to fermions goes as follows. With the conventions of section 2, we define a fermion as a blue dimer, or an empty vertical edge of the even sublattice, shown by a zizag line see below. As we shall see, one complication compared to the honeycomb lattice is that the transfer matrix is invariant with respect to translations of two lattice sites. We are therefore dealing with a two-band problem, in fermion language.

Rectangle                                          Aztec diamond

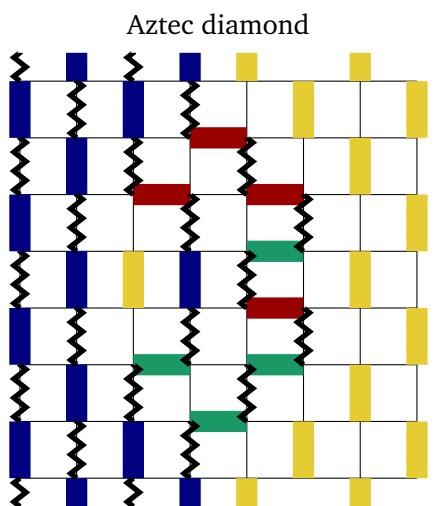

- - - - - - - - - - - - - - - - - - - - - - - - - - - - - - - - - - - - - - - - - - - - - - - - - -

Similar to the honeycomb lattice, we need two transfer matrices $T$ and $T'$, and assume periodic boundary conditions in the horizontal direction.

1. Show $T c_{2j}^{\dagger} T^{-1} = c_{2j}^{\dagger}$, $T c_{2j+1}^{\dagger} T^{-1} = c_{2j}^{\dagger} + c_{2j+1}^{\dagger} + c_{2j+2}^{\dagger}$, and $T |\mathbf{0}\rangle = |\mathbf{0}\rangle$.
2. Show $T' c_{2j}^{\dagger} T'^{-1} = c_{2j-1}^{\dagger} + c_{2j}^{\dagger} + c_{2j+1}^{\dagger}$, $T' c_{2j+1}^{\dagger} T'^{-1} = c_{2j+1}^{\dagger}$, and $T' |\mathbf{0}\rangle = |\mathbf{0}\rangle$.
3. Show

$$(T'T) \begin{pmatrix} a_k^{\dagger} \\ b_k^{\dagger} \end{pmatrix} (T'T)^{-1} = \begin{pmatrix} 1 & 2\cos k \\ 2\cos k & 1 + 4\cos^2 k \end{pmatrix} \begin{pmatrix} a_k^{\dagger} \\ b_k^{\dagger} \end{pmatrix}$$

where $a_k^{\dagger} = \sum_{j=1}^{L/2} e^{\mathrm{i}2kj} c_{2j}^{\dagger}$, $b_k^{\dagger} = \sum_{j=1}^{L/2} e^{\mathrm{i}(2k+1)j} c_{2j+1}^{\dagger}$ and properly quantized momenta $k$

in $[-\pi/2, \pi/2]$.

4. Show $(T'T)f_\pm^\dagger(k)(T'T)^{-1} = \lambda_\pm(k)f_\pm^\dagger(k)$, where $f_+^\dagger(k) = \cos\theta(k)a_k^\dagger + \sin\theta(k)b_k^\dagger$, $f_-^\dagger(k) = -\sin\theta(k)a_k^\dagger + \cos\theta(k)b_k^\dagger$, $\lambda_+(k) = \cot^2\theta(k)$, $\lambda_-(k) = \tan^2\theta(k)$ and

$$\cot 2\theta(k) = \cos k$$

5. Show

$$T'T = \exp\left(-2\sum_k \varepsilon(k)f^\dagger(k)f(k)\right), \tag{113}$$

where the $k$'s are now quantized in $[-\pi, \pi]$,

$$\varepsilon(k) = -\log\left(\cos k + \sqrt{1 + \cos^2 k}\right), \tag{114}$$

and find an expression for $f^\dagger(k)$. [Hint: $\cot^2(\alpha + \pi/2) = \tan^2\alpha$]

---

EXERCISE 4.5     THE AZTEC ARCTIC CIRCLE (II: HYDRODYNAMICS)

The fact that the transfer matrix can be put under the form (113), (114) means we can apply the framework explained in the present chapter. In particular, (91) still holds in the hydrodynamic limit.

6. Show that the hydrodynamic equation

$$x + iy\frac{\sin z}{\sqrt{1 + \cos^2 z}} = R\frac{\cos z}{\sqrt{1 + \cos^2 z}} \tag{115}$$

does implement the correct boundary conditions $\operatorname{Re} z = 0 \ \forall x > R - |y|$ and $\operatorname{Re} z = \pi$ $\forall x < |y| - R$.

7. We introduce the map $z \mapsto \zeta(z) = \arctan[\frac{1}{\sqrt{2}}\tan z]$, initially defined for any $z$ in the strip $\operatorname{Re}(z) \in [-\pi/2, \pi/2]$. We then extend it to $\operatorname{Re} z \in [-\pi, \pi]$ by requiring $\zeta(z \pm \pi) = \zeta(z) \pm \pi$. Show that (115) may be rewritten as

$$x + iy\sin\zeta = \frac{R}{\sqrt{2}}\cos\zeta. \tag{116}$$

Show that (116) has two solutions $\zeta_F$ and $-\zeta_F^*$, with $\operatorname{Re}\zeta_F \in (0, \pi)$, provided $x$ and $y$ satisfy $x^2 + y^2 < R^2/2$.

8. For $y = 0$ and $x^2 < R^2/2$, show that $\zeta_F$ is real, and

$$\langle c_{2j+\sigma}^\dagger c_{2j+\sigma}\rangle = \int_{-z_F}^{z_F} \frac{dk}{\pi}\left(\frac{1}{2} + (-1)^\sigma \frac{\cos 2\theta(k)}{2}\right)$$
$$= \frac{z_F}{\pi} + \frac{(-1)^\sigma}{\pi}\arctan(\sin\zeta_F)$$

where $\sigma = 0, 1$.

9. Compute the probabilities for vertical dimer occupancies along the horizontal line $y = 0$ in the scaling limit.

# 5 Exact lattice calculations

The approach we have taken so far was variational or hydrodynamic: we showed how computing the limit shape boils down to solving PDEs, and found a few cases where this could be done explicitly. It turns out those precise cases can often be treated with other more direct methods. That is, computing explicitely the two point function, and then recover our previous results by a careful asymptotic analysis. This approach is often more technical, but nicely complements hydrodynamics, by confirming its prediction, and also providing more information.

Our aim is to give a flavor how this can be done, on one of the simplest examples. We refer to [51, 76, 77] for similar calculations. We take the transfer matrices from before, and we try to compute the two point function for fermions in the domain wall geometry, where the top and bottom boundary are domain wall-like, with all particles packed to the left of the origin, set at $x = 0$. This is the precise geometry in which the hydrodynamic problem was solved in terms of Hilbert transform (section 4).

Of course as always in a free fermion calculation, if we know the propagator then Wick's theorem allows to reconstruct all higher order correlations. However, as we shall see, computing the two point function is more difficult than in standard condensed matter theory setups.

Let us now specify some notations. We take lattice sites to be half-integers, that is $x \in \mathbb{Z} + 1/2$. We focus on the special case where operators are measured at the same imaginary time $y$, but generalisation is straightforward. We wish to evaluate

$$\langle c_x^\dagger(y,R)c_{x'}(y,R)\rangle = \frac{\langle\psi|e^{-(R-y)H}c_x^\dagger c_{x'}e^{-(R+y)H}|\psi\rangle}{\langle\psi|e^{-2RH}|\psi\rangle}, \tag{117}$$

where expectation values are taken in the domain wall state $|\psi\rangle = \prod_{x<0} c_x^\dagger |0\rangle$. We also have $H = \int \frac{dk}{2\pi}\varepsilon(k)c^\dagger(k)c(k)$, and recall $c_x^\dagger = \int_{-\pi}^{\pi}\frac{dk}{2\pi}e^{-ikx}c^\dagger(k)$, $c^\dagger(k) = \sum_{j\in\mathbb{Z}+1/2}e^{ikj}c_j^\dagger$. Using $e^{\epsilon(k)c^\dagger(k)c(k)}c^\dagger(k)e^{-\epsilon(k)c^\dagger(k)c(k)} = e^{\epsilon(k)}c^\dagger(k)$, this may be rewritten as

$$\langle c_x^\dagger(y,R)c_{x'}(y,R)\rangle = \int_{-\pi}^{\pi}\frac{dk}{2\pi}\int_{-\pi}^{\pi}\frac{dq}{2\pi}e^{-ikx+iqx'}e^{-(R-y)\varepsilon(k)+(R-y)\varepsilon(q)}G_R(k,q), \tag{118}$$

with

$$G_R(k,q) = \frac{\langle c^\dagger(k)c(q)e^{-2RH}\rangle}{\langle e^{-2RH}\rangle}. \tag{119}$$

This remaining term is unusual, and illustrates the extra layer of complexity associated to imaginary time problems –for a real time calculation, just set $y = it$ and $R = 0$. It is very difficult to evaluate (119) in general; however, the special form of the domain wall state in which averages are taken allows for a small miracle.

## 5.1 A nice bosonization trick

Let us work out the case of nearest neighbor hoppings, for which $\varepsilon(k) = -\cos k$. The main player in the calculation will be the operator

$$b = \sum_{x\in\mathbb{Z}+1/2}c_x^\dagger c_{x+1}. \tag{120}$$

Obviously, $H = -(b + b^\dagger)/2$. Think of a finite-size regularisation of the chain, e. g. with sites from $-l$ to $l$. The commutator of $b$ with $b^\dagger$ is given by a telescopic sum, which simplifies to

$$[b,b^\dagger] = c_{-l}^\dagger c_{-l} - c_l^\dagger c_l, \tag{121}$$

which means $\langle\psi|[b, b^\dagger]|\psi\rangle = 1$. Now expand $e^{-2RH}$ in power series. It is easy to check that $\langle\psi|[b, b^\dagger]H^p|\psi\rangle = \langle\psi|H^p|\psi\rangle$ provided $l > p$. Hence for any term in the power series, we can always choose $l$ sufficiently large such that the commutator is scalar.

For any finite $R$ the series is expected to converge quite fast, which means we are allowed to assume $[b, b^\dagger] = 1$ throughout. Hence the operator $b$ is, effectively, a boson. It also happens to annihilate the domain wall state, $b|\psi\rangle = 0$. Now, recall the following formula

$$e^{\alpha(b^\dagger+b)} = e^{\alpha b^\dagger}e^{\alpha b}e^{\frac{\alpha^2}{2}}, \tag{122}$$

which is a special case of the Baker-Campbell-Hausdorff identity[8]. Using this formula both in the numerator and denominator with $\alpha = R$ combined with $e^{\alpha b}|\psi\rangle = |\psi\rangle$, $\langle\psi|e^{\alpha b^\dagger} = \langle\psi|$, yields

$$G_R(k, q) = \langle c^\dagger(k)c(q)e^{Rb^\dagger}\rangle. \tag{123}$$

The last trick is to take derivative. Computing the commutator

$$[c^\dagger(k)c(q), b^\dagger] = \left(e^{-iq} - e^{-ik}\right)c^\dagger(k)c(q), \tag{124}$$

we obtain

$$\partial_R G_R(k, q) = (e^{-iq} - e^{-ik})G_R(k, q). \tag{125}$$

Integrating back we finally obtain

$$G_R(k, q) = e^{R(e^{-iq}-e^{-ik})}G_0(k, q). \tag{126}$$

## 5.2 General dispersion relation

The case of general dispersion relation $\varepsilon(k) = -\sum_{n\geq 1} h_n \cos(nk)$ can be handled in a similar fashion. One introduces the set of operators

$$b_n = \sum_x c_x^\dagger c_{x+n} \quad, \quad n \geq 1, \tag{127}$$

which effectively satisfy the commutation relations

$$[b_n, b_m^\dagger] = n\delta_{nm} \quad, \quad [b_n, b_m] = 0 = [b_n^\dagger, b_m^\dagger]. \tag{128}$$

Using this one can show in a similar fashion

$$\langle e^{2RH}\rangle = e^{(\sum_n nh_n^2)R^2/2}, \tag{129}$$

and

$$G_R(k, q) = e^{R\sum_n h_n(e^{-inq}-e^{-ink})}G_0(k, q). \tag{130}$$

The propagator finally reads

$$\langle c_x^\dagger(y, R)c_{x'}(y, R)\rangle = \int_{-\pi}^{\pi}\frac{dk}{2\pi}\int_{-\pi}^{\pi}\frac{dq}{2\pi}e^{-ikx+iqx'+y(\varepsilon(k)-\varepsilon(q))-iR(\tilde\varepsilon(k)-\tilde\varepsilon(q))}G_0(k, q), \tag{131}$$

where $\tilde\varepsilon(k)$ is the Hilbert transform of $\varepsilon(k)$. Let us insist again that this only holds for expectation values in the domain wall state. All what is left is to compute $G_0(k, q)$. We get from a direct calculation

$$G_0(k, q) = \frac{1}{2i\sin\left(\frac{k-q}{2} - i0^+\right)}. \tag{132}$$

---

[8]A proof of the general formula $e^{s(A+B)} = e^{sA}e^{sB}e^{-(s^2/2)[A,B]}$, valid in case $[A, B]$ commutes with $A$ and $B$ may be obtained by (i) showing that $e^{-sB}Ae^{sB} = A + s[A, B]$ (take derivative) (ii) show that both rhs and lhs of the formula satisfy the same first order differential equation with the same initial data (again take derivative).

## 5.3 Asymptotic analysis

The general method to evaluate the double integral in the limit $R \to \infty$, $x/R$, $x'/R$, $y/R$ fixed is the stationary phase, or steepest descent method. The argument inside the exponential can have very large real and imaginary parts. Writing

$$\theta(k) = kx + iy\varepsilon(k) + R\tilde{\varepsilon}(k), \tag{133}$$

one expects the integral to be dominated, after proper contour deformation, by the region close to the points $k_c$ (resp. $q_c$) where the phase $\theta(k)$ (resp. $-\theta(q)$) becomes stationary. The stationary points are the solution of the equation

$$\theta'(k) = x + iy\varepsilon'(k) + R\tilde{\varepsilon}'(k) = 0, \tag{134}$$

whose solution we denote by $z$ and $-z^*$. This equation is, in fact, identical to (97), which we obtained from the hydrodynamic approach. A full asymptotic analysis falls outside the scope of these lectures. However, let us just mention that what matters is the taylor expansion of the phase around the saddle points, that is the expansion

$$\theta(k) = \theta(z) + \frac{1}{2}\theta''(z)(k-z)^2 + o((k-z)^2). \tag{135}$$

Essentially computing the asymptotics boils down to computing a gaussian integral. The case of coinciding points $x' = x$ is more tricky, since the $k$ and $q$ saddle points might coincide. In that case one has to take into account the pole at $k = q$. See e.g. [78] for the details.

Let us briefly comment on the edge behavior. The arctic curve corresponds to the points where the two solutions (assuming there are two) $z$ and $-z^*$ become equal. This means the second derivative $\theta''(z)$ vanishes, and it becomes necessary to expand to third order

$$\theta(k) = \theta(z) + \frac{1}{6}\theta'''(z)(k-z)^3 + o((k-z)^3). \tag{136}$$

This naturally leads to the Airy kernel, since Airy functions may be alternatively defined as $\mathrm{Ai}(x) = \int_{\mathbb{R}+i0^+} \frac{dk}{2\pi} e^{ikx+ik^3/3}$. The subject would require longer exposition, but we have illustrated the two equivalent through which the Airy kernel emerges. Either with fermions in a potential that becomes linear in a Hamiltonian point of view, or through coalescence of two saddle points.

EXERCISE 5.1    BOSONIZING TOEPLITZ DETERMINANTS    [76, 79, 80]

A semi-infinite Toeplitz matrix is a matrix $T = (T_{ij})_{i,j \in \mathbb{N}}$ whose elements depend only on the difference $i - j$, $T_{ij} = g_{i-j}$. It is convenient to interpret the $g_l$ as Fourier coefficients of a periodic function, sometimes called symbol:

$$g(k) = \sum_{l \in \mathbb{Z}} e^{ikl} g_l \qquad , \qquad g_l = \int_{-\pi}^{\pi} \frac{dk}{2\pi} e^{-ikl} g(k) \qquad (137)$$

We assume that $g$ is sufficiently smooth, has a well-defined logarithm which we denote by $\varepsilon(k) = \log g(k)$, and also that $\int_{-\pi}^{\pi} \frac{dk}{2\pi} \varepsilon(k) = 0$. Consider the free fermions Hamiltonian $H = \int_{-\pi}^{\pi} \frac{dk}{2\pi} \varepsilon(k) c^\dagger(k) c(k)$, with conventions (183), which reads $H = \sum_{j \in \mathbb{Z}} \sum_{p>0} \varepsilon_p (c^\dagger_{j+p} c_j + h.c)$ in real space, where the $\varepsilon_p$ are the Fourier coefficients of $\varepsilon(k)$. We introduce a similar domain wall state $|\phi\rangle = \prod_{j=0}^{\infty} c^\dagger_j |0\rangle$ as in the text, where notice the fermions are now located on nonnegative integers.

----

1. Show using bosonization that $\langle \phi | e^H | \phi \rangle = \exp\left( \frac{1}{2} \sum_{p=1}^{\infty} p \varepsilon_p \varepsilon_{-p} \right)$.

2. Show that $T_{ij} = \langle 0 | c_j e^H c^\dagger_i | 0 \rangle$ and $(T^{-1})_{ij} = \frac{\langle \phi | c^\dagger_i e^H c_j | \phi \rangle}{\langle \phi | e^H | \phi \rangle}$.

3. Show using bosonization that

$$(T^{-1})_{ij} = \int_{-\pi}^{\pi} \frac{dk}{2\pi} \int_{-\pi}^{\pi} \frac{dq}{2\pi} e^{-i(ki-qj)} g_+^{-1}(k) g_-^{-1}(q) \frac{1}{1 - e^{-i(k-q-i0^+)}},$$

where the $g_{\pm}(k) = \exp\left( \sum_{\pm n > 0} \varepsilon_n e^{ikn} \right)$ are the Wiener-Hopf factors of $g(k)$.

----

We now look at a finite $2N \times 2N$ truncation of $T$, which we note $T_N$. We want to evaluate $\det T_N = \det_{0 \le i, j \le 2N-1} (g_{i-j})$ in the limit $N \to \infty$. For this purpose, we introduce the state $|\psi_N\rangle = \prod_{|x|<N} c^\dagger_x |0\rangle$ where the sites are now put on the half-integer line $x \in \mathbb{Z} + 1/2$.

4. Show using Wick's theorem that $\det T_N = \langle \psi_N | e^H | \psi_N \rangle$.

Let us introduce the 'right modes' $r_n = \sum_{x>0} c^\dagger_x c_{x+n}$ as well as the 'left modes' $l^\dagger_n = \sum_{x<0} c^\dagger_x c_{x+n}$ for $n \in \mathbb{N}^*$.

5. Under which condition on $n, m, N$ do we have $[r_n, b_m] = 0$? $[r_n, r^\dagger_m] = n\delta_{nm}$? $[l_n, l^\dagger_m] = n\delta_{nm}$?

6. Bosonize and show that when $N \to \infty$ the determinant should converge to

$$\lim_{N \to \infty} \det T_N = \exp\left( \sum_{p=1}^{\infty} p \varepsilon_p \varepsilon_{-p} \right)$$

provided the series inside the exponential converges sufficiently fast. This result was first found by Onsager-Kaufman [79], and proved by Szegö [80] shortly thereafter (both used different techniques).

7. You are Onsager and Kaufman, and you just realized that the spin-spin correlations $\langle \sigma_{0,0} \sigma_{n,n} \rangle$ along the diagonal of the classical isotropic Ising model on $\mathbb{Z}^2$ are given by a $n \times n$ Toeplitz determinant with symbol $g(k)^2 = \frac{1 - \alpha e^{-ik}}{1 - \alpha e^{ik}}$, where $1/\alpha = \sinh^2(\beta J)$. This holds below the critical temperature, $J\beta > J\beta_c = \text{arcsinh } 1$. What is the magnetisation exponent of the 2d Ising model?

# 6 Conclusion and related problems

We finish with a discussion of a few selected topics that go beyond the lectures, but still fit well with the spirit of the notes. We first examine the effects of interactions (section 6), how this (does not) affect the edge behavior in section 6.2, and finally explore the intricacies of the Wick rotation, section 6.3.

## 6.1 Interactions

For interacting systems, i.e. systems that cannot be mapped onto free fermions, the logic of the present notes still applies. The difference is that no exact formula for the free energy exists in general anymore. There are however deformations of the dimer model for which some analytical progress is possible. Those models are called *integrable*.

A discussion of all the intricacies of integrable models falls well outside the scope of the present notes, see [53, 54, 81] for reviews. We consider the case of the six vertex model, or even plaquettes interacting dimers, as explained in section 1. We parametrize the interaction term as $e^\lambda = 1 - \Delta$ or $e^\lambda = 1 - \cos\gamma$, depending on convenience. The only result that we need here is the following fact: in the same way that the free energy at the free fermion point could be determined from a simple ground state energy with some current,

$$F(r,s) = \sum_{|k|<k_F} \varepsilon(k+i\nu) - \nu s, \qquad (138)$$

where $\nu$ and $k_F$ are determined from $r,s$, a similar expressions holds true for half-interacting dimers (or the six vertex model). Namely, the free energy is determined from the biggest eigenvalue of the transfer matrix (with appropriate particle number and current). The latter is given by [40, 82]

$$\Lambda = e^{-L\nu} \prod_{j=1}^{N} \frac{\sinh(\lambda_j + i\gamma)}{\sinh\lambda_j} + e^{L\nu} \prod_{j=1}^{N} \frac{\sinh(\lambda_j - i\gamma)}{\sinh\lambda_j}. \qquad (139)$$

The $\lambda_j$ play a role similar to momenta for free fermions, and $N/L$ controls $r$. The big difference is that the quantization condition is much more complicated. It is given by a set of equations

$$\left[ e^{-2\nu} \frac{\sinh(\lambda_i + i\gamma/2)}{\sinh(\lambda_i - i\gamma/2)} \right]^L = \prod_{j\neq i}^{N} \frac{\sinh(\lambda_i - \lambda_j + i\gamma)}{\sinh(\lambda_i - \lambda_j - i\gamma)}, \qquad (140)$$

called Bethe equations[9]. The free case corresponds to $\gamma = \pi/2$ for which the rhs simplifies to 1, and the $\lambda_k$ can be obtained explicitly. The reader can easily imagine that solving the Bethe equations is in general extremely difficult. The fact that (away from the free fermion point) the rhs is a complicated product over the positions of the particles has an important physical consequence, which usually goes under the name *dressing*: changing the number of particles ($N$) affects all the rapidities, as illustrated in figure 18.

Dressing severely complicates asymptotic analysis, since any $\sum_j f(\lambda_j)$ becomes in the thermodynamic limit $\int f(\lambda)\rho(\lambda)d\lambda$, where $\rho$ is the density of Bethe roots. Now comes the big problem: the root density is only known in the case of zero current $\nu = 0$, in which case it is a solution to a linear integral equation over a segment of the real line [53]. In the case of nonzero current these aspects have been investigated numerically in Ref. [40], and one finds that the Bethe roots condense on some non trivial curve in the complex plane.

---

[9]The standard Bethe equations for the six vertex model, or its Hamiltonian limit the spin-1/2 XXZ chain, correspond to the case $\nu = 0$. Here we need a slight generalisation to induce some imaginary current.

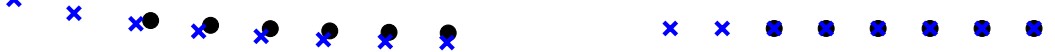

**Figure 18: Dressing**

Illustration of dressing. Left: Case $\Delta = 0.6$, $L = 32$. Black circles are half the Bethe roots for $N = 12$, red crosses for $N = 16$. Right: same for the free case $\Delta = 0$ or $\gamma = \pi/2$.

A particular case that can be treated exactly is the so-called five vertex model, obtained from the six vertex model by setting one of the vertex weights to zero –it can also be seen as an (half-plaquettes) interacting version of the honeycomb dimer model [83]. This model is related to stochastic processes such as TASEP. The exact free energy can be computed, and limit shapes are also parametrized by analytic functions [55]. This is to date one of the most complicated model in which the variational/hydrodynamic program has been applied. This model is also the only one for which one can show that the Luttinger parameter $K$ is not constant.

For the full six vertex model the hydrodynamic program has not been completed, the main bottleneck being determining the exact curve on which the roots densify. However, the arctic curve has been determined analytically, using the lattice approach. In a series of papers [84, 85], Colomo and Pronko, and Colomo-Pronko-Zinn-Justin managed to compute exactly the lattice emptiness formation probability, which gives the distribution of the last particle. They managed to determine the precise location where this probability goes to zero in the thermodynamic limit. This location coincides with the arctic curve. Except at special values where $\gamma/\pi$ is a rational number, this curve is non algebraic. Later, an attractive *tangent method* was also introduced to get the arctic curve in a slightly simpler way [86]. This method was also recently used to provide a proof [87] of the Colomo-Pronko formula for the curve in the special case $\Delta = 1/2$, the so-called combinatorial point.

## 6.2 Tracy-Widom at the edge

Tracy-Widom scaling is, in fact, also expected at the edge, for the following simple physical reason: near the edge the particle (or hole) density goes to zero, hence particles (holes) are diluted. For local interaction such as the plaquette terms we discussed previously, it is reasonable to assume that interactions become weaker and weaker. Hence particles become effectively free near the edge, and we expect the arguments presented in section 4.6 to hold, and T-W scaling at the edge. One can check numerically that this is the case: in figure 19 we show Monte Carlo simulations in interacting dimers and the six vertex model. We compute the lattice emptiness formation probability numerically, rescale appropriately, and compare to the T-W distribution. As can be seen the agreement is quite good. We also compute the skewness of the discrete distribution, and compare it to T-W, with excellent (and improving for larger $L$) agreement. Other similar checks in inhomogeneous quantum chains can also be found in [75], or with anharmonic chains in thermal equilibrium [88].

Showing this analytically in some generality in integrable models is not easy, despite the fact that the lattice emptiness formation probability is known exactly in the six vertex model. Note that the argument above does not assume integrability. However, the dilution argument can nicely be illustrated in Bethe-Ansatz integrable models, such as six vertex. Indeed looking back at the Bethe equations (140), the edge diluted limit corresponds to $N \ll L$, for which the rhs can be considered a constant. Hence in this limit dressing disappears, and we are back to free fermions.

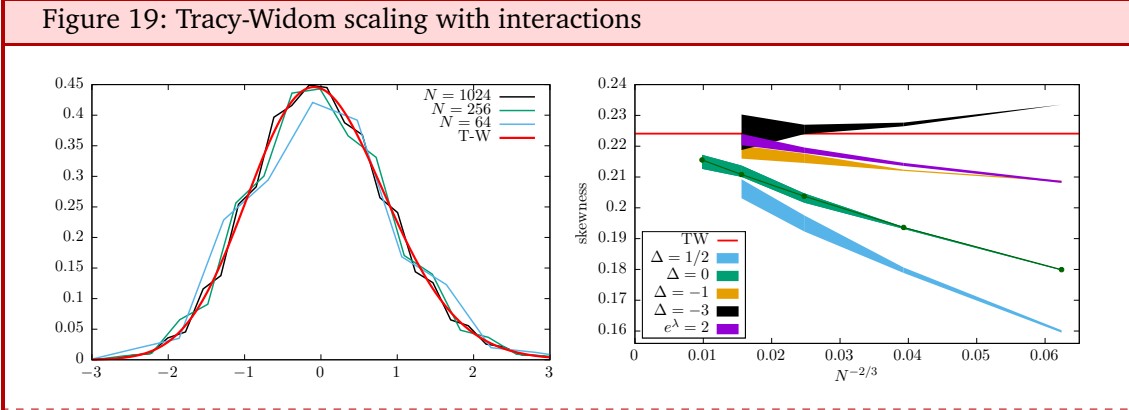

**Figure 19: Tracy-Widom scaling with interactions**

Numerical check of Tracy-Widom scaling for interacting dimers and half-plaquette interacting dimers (aka six vertex model). Left: rescaled distribution for interacting dimers ($e^\lambda = 2$) and $N = 64, 256, 512$, compared to the centered T-W distribution (thick red line). Right: skewness $\text{sk} = \mathbb{E}[(s - \mathbb{E}[s])^3]$ as a function of $N^{-2/3}$, for both interacting dimers, and six vertex model in which case we show the corresponding value of $\Delta$. This is compared to the T-W value $\text{sk} \simeq 0.22408$. The thickness of the lines are the Monte Carlo error bars. The exact lattice skewness is also shown for comparison at the free fermions point $\Delta = 0$ (bullets).

On the rigorous side, stochastics processes such are ASEP appear to be more tractable. For example, a proof of T-W scaling is known for ASEP with step initial conditions [89].

## 6.3 Wick rotation and inhomogeneous quantum quenches

We have seen that the expectation value of an observable $O_x$ in imaginary time is given by

$$\langle O_x \rangle = \frac{\langle \psi | e^{-(R+y)H} O_x e^{-(R-y)H} | \psi \rangle}{\langle \psi | e^{-2RH} | \psi \rangle}, \tag{141}$$

with either $R$ and $y$ discrete (dimers models, etc) or $R$ and $y$ continuous (XX chain in imaginary time). Our starting point will be the latter case. The reader interested in quantum models might have spotted a similarity between the previous expression and regular time evolution steming from the Schrödinger equation. Indeed, for a quantum system prepared in a state $|\psi\rangle$, and let evolve in time with the Hamiltonian $H$, the wave function at time $t$ is $|\psi(t)\rangle = e^{-iHt} |\psi\rangle$, and the expectation value of $O_x$ at time $t$ becomes

$$\langle O_x(t) \rangle = \langle \psi(t) | O_x | \psi(t) \rangle \tag{142}$$

$$= \langle \psi | e^{iHt} O_x e^{-iHt} | \psi \rangle . \tag{143}$$

Formally the real time evolution may be recovered from setting $y = -it$ in (141) and taking the limit $R \to 0^+$. This procedure is the famous *Wick rotation*.

This observation can be useful for two reasons.

a) First, exact calculation in the spirit of section 5 or using integrability techniques are quite algebraic in nature. For this reason they are often valid for any value of $R, y \in \mathbb{C}$, which means the Wick rotation is perfectly justified for any finite time $t$. We can use this to derive exact highly nontrivial expressions for out of equilibrium quantities in a few selected case.

For example, the partition function $Z(R) = \langle \psi | e^{-RH} | \psi \rangle$ of the six vertex model with domain wall boundary conditions is known exactly from the work of Korepin [90] and Izergin [91] (see also [92]). We can then take the Hamiltonian limit, perform the Wick rotation, and get an exact expression [93] for the amplitude $\langle \psi | e^{itH_{XXZ}} | \psi \rangle$, where $H_{XXZ}$ is the Hamiltonian of the XXZ spin chain, an integrable generalisation of the XX chain. The modulus square of the amplitude is called return probability, or Loschmidt echo. This exact result is difficult to get from more direct approaches [94], which makes this method worthwhile.

b) At a more speculative level, it is tempting to try and Wick-rotate results already in the thermodynamic limit, following [95]. Of course, there is no mathematical justification for this. The statement that this can lead to wrong results is called *Stokes phenomenon*. Let us go back to our quench from a domain wall state. Hydrodynamics ideas can also be applied to out of equilibrium quantum integrable sytems, as was established recently [96, 97]. This subject goes under the name *generalized hydrodynamics*. For a quench from a domain wall state a small miracle occurs, and it is possible to get the density profile exactly [98]. What is nice is that the Wick rotation of the arctic curve gives back precisely the location of the front, the simplest example being the free fermion point arctic circle $x^2 + y^2 = R^2$ which becomes $x = \pm t$ after Wick rotation.

This does not work for all observables, though. For example the return probability or the density profile have the crazy property of being highly non regular as a function of $\Delta$ in the thermodynamic limit, which is clearly not the case in the original statistical problem.

# Acknowledgements

I am grateful to many researchers for discussions and collaboration on various topics very much related to the present lectures. Among those are Jérémie Bouttier, Filippo Colomo, Jérôme Dubail, Paul Fendley, Christophe Garban, Grégoire Misguich, Vincent Pasquier, Fabio Toninelli and Jacopo Viti. I thank Saverio Bocini for a careful reading of these notes, suggesting various improvements, and helping correct many misprints. I also thank Alejandro Caicedo for working out the intricacies of exercises 4.4 and 4.5.

**Funding information.** The ANR-18-CE40-0033 grant ('DIMERS') is warmly acknowledged.

# A  Self-contained reminder on free fermions techniques

## A.1  An explicit construction of lattice fermions

**Two level system.**    The two level system is obviously the most important genuine quantum system ever, so let us start from this one. We consider the Hilbert space $\mathcal{H} \simeq \mathbb{C}^2$, and take the most down to earth approach, which is to work with two by two matrices ($\dim \mathcal{H} = 2$). We choose the two basis vectors $|0\rangle = \begin{pmatrix} 0 \\ 1 \end{pmatrix}$ and $|1\rangle = \begin{pmatrix} 1 \\ 0 \end{pmatrix}$. $|1\rangle$ is interpreted as the presence of a particle, $|0\rangle$ as the absence of a particle (vacuum state). Pure states (or wave functions) are of the form

$$|\psi\rangle = \alpha |0\rangle + \beta |1\rangle \qquad , \qquad (\alpha, \beta) \in \mathbb{C}^2. \tag{144}$$

Now let us introduce the two main heroes,

$$c = \begin{pmatrix} 0 & 0 \\ 1 & 0 \end{pmatrix} \tag{145}$$

and

$$c^\dagger = \begin{pmatrix} 0 & 1 \\ 0 & 0 \end{pmatrix}. \tag{146}$$

$\dagger$ denotes hermitian conjugate. We use bra/ket notations, e.g. the bra $\langle 0| = (|0\rangle)^\dagger = (0\ 1)$ is a line vector. We have $c^\dagger |0\rangle = |1\rangle$, $c|1\rangle = |0\rangle$. Since $c|1\rangle = |0\rangle$, $c$ destroys the particle, so we call it the *annihilation operator*. $c^\dagger |0\rangle = |1\rangle$ creates a particle from the vacuum, so $c^\dagger$ is the *creation operator*. We also have $c^\dagger |1\rangle = c|0\rangle = \begin{pmatrix} 0 \\ 0 \end{pmatrix} = 0$, so that it is not possible to create two particles, or destroy a non existent particle. The zero vector is not to be confused with the vacuum. The rightmost hand side of the previous equation involves a slight abuse of notations; in the following we keep on writing 0 any vector/matrix that has all elements equal to zero.

Note also $c^\dagger c = \begin{pmatrix} 1 & 0 \\ 0 & 0 \end{pmatrix}$, and $c^\dagger c + c c^\dagger = I_2 = \begin{pmatrix} 1 & 0 \\ 0 & 1 \end{pmatrix}$, which will be useful in the following. Obviously, any matrix in $M_2(\mathbb{C})$ can be written as a linear combination of $c, c^\dagger, c^\dagger c$ and $c c^\dagger$.

**A collection of $L$ two level systems.** We now consider the Hilbert space $\mathcal{H} \simeq (\mathbb{C}^2)^{\otimes L}$, where $L$ is an integer $\geq 2$. $\mathcal{H}$ has dimension $\dim \mathcal{H} = 2^L$. We want a set of (Dirac) fermionic operators, that is, a set of $2^L \times 2^L$ matrices $c_i, c_i^\dagger$ for $i = 1, \ldots, L$ that satisfy

$$c_i c_j^\dagger + c_j^\dagger c_i = \delta_{ij} I \tag{147}$$
$$c_i c_j = -c_j c_i, \tag{148}$$

where $I = I_2 \otimes \ldots \otimes I_2$ is the identity operator. $\otimes$ denotes tensor product, given by

$$\begin{pmatrix} a_0 & b_0 \\ c_0 & d_0 \end{pmatrix} \otimes \begin{pmatrix} a_1 & b_1 \\ c_1 & d_1 \end{pmatrix} = \begin{pmatrix} a_0 a_1 & a_0 b_1 & b_0 a_1 & b_0 b_1 \\ a_0 c_1 & a_0 d_1 & b_0 c_1 & b_0 d_1 \\ c_0 a_1 & c_0 b_1 & d_0 a_1 & d_0 b_1 \\ c_0 c_1 & c_0 d_1 & d_0 c_1 & d_0 d_1 \end{pmatrix}. \tag{149}$$

The fact that it satisfies

$$(A \otimes B)(C \otimes D) = (AC) \otimes (BD) \tag{150}$$

is more important than the explicit formula (149).

The relations (147) and (148) are called canonical anticommutation relations (CAR), since they involve the anticommutator $\{A, B\} = AB + BA$, instead of the commutator $[A, B] = AB - BA$.

An explicit construction, due to Jordan and Wigner [99] [10] is given by:

$$c_1^\dagger = c^\dagger \otimes \underbrace{I_2 \otimes \ldots \otimes I_2}_{N-1 \text{ times}} \tag{151}$$

$$\vdots$$

$$c_k^\dagger = \underbrace{\begin{pmatrix} -1 & 0 \\ 0 & 1 \end{pmatrix} \otimes \ldots \otimes \begin{pmatrix} -1 & 0 \\ 0 & 1 \end{pmatrix}}_{k-1 \text{ times}} \otimes c^\dagger \otimes \underbrace{I_2 \otimes \ldots \otimes I_2}_{N-k \text{ times}} \tag{152}$$

$$\vdots$$

$$c_N^\dagger = \underbrace{\begin{pmatrix} -1 & 0 \\ 0 & 1 \end{pmatrix} \otimes \ldots \otimes \begin{pmatrix} -1 & 0 \\ 0 & 1 \end{pmatrix}}_{N-1 \text{ times}} \otimes c^\dagger. \tag{153}$$

The chain of $k-1$ tensor products of $(-1)^{c^\dagger c} = e^{i\pi c^\dagger c}$ in (152) is called a *Jordan-Wigner string*. One can readily check that the CAR (147), (148) are satisfied, using (150) lots of times.

Of course, in practice, one often just uses the CAR without caring about an explicit representation. However, in the context of classical statistical mechanics, the above explicit construction turns out to be very useful.

## A.2  Summary of useful fermions properties

Starting from now, we start droping the identity matrix in the equations, and simply treat it as a scalar. The CAR now read

$$c_i^\dagger c_j = \delta_{ij} - c_j c_i^\dagger \quad , \quad c_i c_j = -c_j c_i \quad , \quad c_i^\dagger c_j^\dagger = -c_j^\dagger c_i^\dagger. \tag{154}$$

In particular $c_i c_i = 0 = c_i^\dagger c_i^\dagger$. The following properties all follow rather straightforwardly from (154) and the existence of a vacuum state $|\mathbf{0}\rangle = |0\rangle \otimes \ldots \otimes |0\rangle$ annihilated by all $c_i$, $i = 1, \ldots, L$:

- $\langle \mathbf{0} |$ is annihilated by all $c_i^\dagger$, $\langle \mathbf{0} | c_i^\dagger = 0$, $\forall i \in \{1, \ldots, L\}$.

- Commutation relations for quadratic forms:

$$[c_i^\dagger c_j, c_k^\dagger c_l] = \delta_{jk} c_i^\dagger c_l - \delta_{il} c_k^\dagger c_j. \tag{155}$$

  In particular,

$$[c_i^\dagger c_i, c_k^\dagger c_k] = 0. \tag{156}$$

- Exponentiation [follows from the fact that $c_i^\dagger c_i$ is idempotent, $(c_i^\dagger c_i)^2 = c_i^\dagger c_i$]:

$$\exp\left(\tau c_i^\dagger c_i\right) = 1 + (e^\tau - 1) c_i^\dagger c_i. \tag{157}$$

- Time evolution [take derivative]:

$$e^{\tau c_i^\dagger c_i} c_j^\dagger e^{-\tau c_i^\dagger c_i} = e^{\tau \delta_{ij}} c_j^\dagger. \tag{158}$$

---

[10] This result is often used when studying quantum spin chains, which are modeled using Pauli matrices. The construction goes $\sigma_j^\alpha = I_2 \otimes \ldots \otimes \sigma^\alpha \otimes I_2 \ldots \otimes I_2$, for $\alpha = \text{x}, \text{y}, \text{z}$, where $\sigma^x = c^\dagger + c$, $\sigma^y = -ic^\dagger + ic$, $\sigma^z = 2c^\dagger c - I_2$ are the Pauli matrices. The "Pauli matrices acting on site $j$" are related to fermions through the *Jordan-Wigner transformation* $\sigma_j^z = 2c_j^\dagger c_j - I$, $\sigma_j^x + i\sigma_j^y = 2c_j^\dagger \prod_{l=1}^{j-1} \left(I - 2c_l^\dagger c_l\right) = 2c_j^\dagger (-1)^{\sum_{l=1}^{j-1} c_j^\dagger c_j}$.

### A.3  How to diagonalize a free fermions Hamiltonian?

A free (lattice) fermions Hamiltonian is a $2^L \times 2^L$ matrix that is quadratic in the fermions creation and annihilation operators (we assume hermiticity here, which is not necessary, strictly speaking):

$$H = \sum_{i,j=1}^{L} \left( A_{ij} c_i^\dagger c_j + B_{ij} c_i^\dagger c_j^\dagger + B_{ij}^* c_j c_i \right), \tag{159}$$

where $A$ and $B$ are $L \times L$ matrices ($A$ is a Hermitian). This is of course a specific class of Hamiltonians, since we have at most $2L^2$ free parameters while the Hilbert space size is $2^L$. In this context, free really means quadratic in the fermion operators.

In the following we explain how to diagonalise $H$ in the special case $B = 0$, for simplicity. The procedure described below can be generalized to treat cases where $B$ is a non zero matrix. Hamiltonians of the form

$$H = \sum_{i,j=1}^{L} A_{ij} c_i^\dagger c_j \tag{160}$$

conserve the number of particles, which means applying it on $n$-particle states $c_{i_1}^\dagger \dots c_{i_n}^\dagger |\mathbf{0}\rangle$ returns a sum over $n$ particle states (any fermion destroyed by $c_j$ is immediately created back by $c_i^\dagger$). $A$ is a hermitian $L \times L$ matrix, so can be diagonalized in an orthonormal basis. The corresponding eigenvalue equations read (assume no multiplicities for simplicity)

$$\sum_{j=1}^{L} A_{ij} u_{jk} = \epsilon_k u_{ik} \quad , \quad k = 1, \dots, L. \tag{161}$$

The eigenvalues are the $\epsilon_k$ and the $u_{jk}$ are orthonormal, meaning $\sum_{j=1}^{L} u_{jk}^* u_{jq} = \delta_{kq}$. Now introduce a new set of fermions as

$$f_k^\dagger = \sum_{j=1}^{L} u_{jk} c_j^\dagger \quad , \quad k = 1, \dots, L \quad , \quad f_k = (f_k^\dagger)^\dagger. \tag{162}$$

Then it is easy to show $\{f_k, f_q^\dagger\} = f_k f_q^\dagger + f_q^\dagger f_k = \delta_{kq}$ and $\{f_k, f_q\} = f_k f_q + f_q f_k = 0$, so the new set of operators also obey the CAR. In terms of these the Hamiltonian reads

$$H = \sum_{k=1}^{L} \epsilon_k f_k^\dagger f_k. \tag{163}$$

Obtaining the spectrum becomes quite easy now. Obviously $H |\mathbf{0}\rangle = 0$. Using the anticommutation relations, $H f_k^\dagger |\mathbf{0}\rangle = \epsilon_k f_k^\dagger |\mathbf{0}\rangle$. By induction, we obtain

$$H f_{k_1}^\dagger f_{k_2}^\dagger \dots f_{k_n}^\dagger |\mathbf{0}\rangle = (\epsilon_{k_1} + \dots + \epsilon_{k_n}) f_{k_1}^\dagger f_{k_2}^\dagger \dots f_{k_n}^\dagger |\mathbf{0}\rangle. \tag{164}$$

To get a nonzero eigenvector, the $k_i$ have to be pairwise distincts. Also, any permutation of the $k_i$s gives back the same eigenvector up to a sign. Hence the spectrum is

$$\sum_{i=1}^{n} \epsilon_{k_i} \quad , \quad \{k_1, \dots, k_n\} \text{ subset of } \{1, \dots, L\}.$$

There are $\binom{L}{n}$ linearly independent eigenvectors in the sector with $n$ particles, so the total number of eigenvalues is $\sum_{n=0}^{L} \binom{L}{n} = 2^L = \dim \mathcal{H}$, as should be.

An eigenstate with smallest eigenvalue is obtained by choosing all single particle energies $\epsilon_k$ that are negative, $|\Omega\rangle = \prod_{k, \epsilon_k < 0} f_k^\dagger |\mathbf{0}\rangle$ (irrespective of the order in which the product is taken).

### A.4 More elaborate properties

Here is a collection of results that are very useful in practice. Almost all free fermions calculations make use of several of those at some point. We start with the most famous one.

a) Wick's theorem [100]. Let the $f_j$ be linear combinations of the $c_i, c_i^\dagger$ for $j \in \{1, \ldots, 2n\}$, and $H$ be a free fermion Hamiltonian. Then the thermal average

$$\langle f_1 \ldots f_{2n} \rangle_\beta = \frac{\mathrm{Tr}\left[f_1 \ldots f_{2n} e^{-\beta H}\right]}{\mathrm{Tr}\left[e^{-\beta H}\right]} \tag{165}$$

may be expressed as

$$\langle f_1 \ldots f_{2n} \rangle_\beta = \Pf_{1 \le i,j \le 2n}\left(\left\langle \mathcal{T}[f_i f_j]\right\rangle_\beta\right), \tag{166}$$

where $\mathcal{T}[f_i f_j]$ equals $f_i f_j$ if $i < j$, 0 if $i = j$, and $-f_j f_i$ if $i > j$. Pf is the Pfaffian. For an antisymmetric matrix $A = (A_{ij})_{1 \le i,j \le 2n}$, it is defined as

$$\Pf A = \frac{1}{2^n n!} \sum_{\sigma \in S_{2n}} (-1)^\sigma A_{\sigma(1),\sigma(2)} \ldots A_{\sigma(2n-1)\sigma(2n)}, \tag{167}$$

where the sum runs over all permutations $\sigma$ of $\{1, \ldots, 2n\}$, and $(-1)^\sigma$ is the signature of the permutation. The Pfaffian can be shown to be a square root of the determinant.

The factor $1/(2^n n!)$ may also be removed by imposing the ordering $\sigma(2i) < \sigma(2i+1)$, and $\sigma(2i) < \sigma(2j)$ for $j > i$. As an example, the theorem yields

$$\langle f_1 f_2 f_3 f_4 \rangle_\beta = \langle f_1 f_2 \rangle_\beta \langle f_3 f_4 \rangle_\beta - \langle f_1 f_3 \rangle_\beta \langle f_2 f_4 \rangle_\beta + \langle f_1 f_4 \rangle_\beta \langle f_2 f_3 \rangle_\beta. \tag{168}$$

We refer to [101] for a proof of (166). Note that as a particular case, expectation values in any ground state may be obtained by taking the limit $\beta \to \infty$.

b) Wick's theorem (no pairings). We consider the special case where $H$ is of the form (160), while for $j = 1, \ldots, n$ the $f_j$ are linear combinations of the $c_i^\dagger$ only, and the $f_{j+n} = g_j$ are linear combinations of the $c_j$ only. Then

$$\langle f_1 \ldots f_{2n} \rangle_\beta = \det_{1 \le i,j \le n}\left(\langle f_i g_j \rangle_\beta\right). \tag{169}$$

We will need only this particular case in the limit $\beta \to \infty$ in these notes.

c) Trace of exponential.

$$\mathrm{Tr}\, e^{-\beta \sum_{i,j} P_{ij} c_i^\dagger c_j} = \det(1 + e^{-\beta P}). \tag{170}$$

*Proof*: diagonalize the quadratic form in the exponential.

d) General time evolution.

$$e^{\sum_{i,j} P_{ij} c_i^\dagger c_j} c_l^\dagger e^{-\sum_{i,j} P_{ij} c_i^\dagger c_j} = \sum_m (e^P)_{ml} c_m^\dagger, \tag{171}$$

$$e^{\sum_{i,j} P_{ij} c_i^\dagger c_j} c_l e^{-\sum_{i,j} P_{ij} c_i^\dagger c_j} = \sum_m (e^{-P})_{lm} c_m. \tag{172}$$

*Proof*: diagonalize the quadratic form in the exponentials.

*e)* Product of exponentials.

$$e^{\sum_{i,j} P_{ij} c_i^\dagger c_j} e^{\sum_{i,j} Q_{ij} c_i^\dagger c_j} = e^{\sum_{i,j} \log(e^P e^Q)_{ij} c_i^\dagger c_j}. \tag{173}$$

*Proof*: use the Baker-Campbell-Hausdorff formula.

*f)* Average of an exponential. Let $P = (P_{ij})_{1 \le i,j \le L}$ be a $L \times L$ matrix. Then in any state where Wick's theorem applies we have

$$\left\langle e^{\sum_{i,j} P_{ij} c_i^\dagger c_j} \right\rangle = \det(I + (e^P - I)C). \tag{174}$$

$C$ is the $L \times L$ matrix with elements $C_{ij} = \langle c_i^\dagger c_j \rangle$, $I$ the $L \times L$ identity. Of course, it is also possible to combine (174) with (173) to compute the average of a product of exponentials.

The formula gives the full counting statistics as byproduct: for any subset $A$ of $\{1, 2, \ldots, L\}$:

$$\left\langle e^{\lambda \sum_{j \in A} c_j^\dagger c_j} \right\rangle = \det_{i,j \in A} \left( \delta_{ij} + [e^\lambda - 1] \langle c_i^\dagger c_j \rangle \right). \tag{175}$$

*Proof*: just pick $P_{ij} = \lambda \delta_{ij} \delta_{i \in A}$ in (174) and check that only the block $i, j \in A$ contributes to the determinant.

*Proof of (174):* Introduce a new set of fermions $d_k, d_k^\dagger$ that diagonalise the quadratic form in the exponential on the lhs, as in section A.3. Denote by $\epsilon_k$ the corresponding eigenvalues of $P$. The $d_k, d_k^\dagger$ also satisfy the CAR. Then

$$\left\langle e^{\sum_{i,j=1}^L P_{ij} c_i^\dagger c_j} \right\rangle = \left\langle e^{\sum_{k=1}^L \epsilon_k d_k^\dagger d_k} \right\rangle, \tag{176}$$

$$= \left\langle \prod_{k=1}^L e^{\epsilon_k d_k^\dagger d_k} \right\rangle, \tag{177}$$

$$= \left\langle \prod_{k=1}^L \left( 1 + [e^{\epsilon_k} - 1] d_k^\dagger d_k \right) \right\rangle, \tag{178}$$

$$= \left\langle \sum_{n=0}^L \sum_{k_1 < \ldots < k_n} \prod_{\alpha=1}^n (e^{\epsilon_{k_\alpha}} - 1) d_{k_\alpha}^\dagger d_{k_\alpha} \right\rangle, \tag{179}$$

$$= \sum_{n=0}^L \sum_{k_1 < \ldots < k_n} \det_{1 \le \alpha, \beta \le n} \left( [e^{\epsilon_{k_\alpha}} - 1] \langle d_{k_\alpha}^\dagger d_{k_\beta} \rangle \right), \tag{180}$$

$$= \det_{1 \le k, q \le L} \left( \delta_{kq} + [e^{\epsilon_k} - 1] \langle d_k^\dagger d_q \rangle \right). \tag{181}$$

We have used in succession (156), (157), expanded the product over $k$, Wick's theorem, and recognised the Cauchy-Binet identity. Writing $P = U \Delta U^\dagger$, where $\Delta = \text{diag}(\epsilon_1, \ldots, \epsilon_L)$, (181) reads

$$\left\langle e^{\sum_{i,j=1}^L P_{ij} c_i^\dagger c_j} \right\rangle = \det(I + (e^\Delta - I) U^\dagger C U). \tag{182}$$

Finally, multiplying by $U$ on the left and $U^\dagger$ on the right does not change the determinant, and gives (174).

### A.5 Infinite lattice or continuum limit

So far we have introduced lattice fermions as $2^L \times 2^L$ matrices, where $L$ is the number of lattice sites. It is however very useful to consider generalisations where the matrices become infinite or operators in the continuum. The case of the infinite lattice, e. g. $\mathbb{Z}$ can easily be dealt with by considering a finite lattice $j \in \{-l, -l + 1, \dots, l\}$ computing observables $\langle O \rangle_l$ and then taking $l \to \infty$.

We often make use of the following notation

$$c^\dagger(k) = \sum_{x \in \mathbb{Z}} e^{ikx} c_x^\dagger \qquad , \qquad c_x^\dagger = \int_{-\pi}^{\pi} \frac{dk}{2\pi} e^{-ikx} c^\dagger(k) \tag{183}$$

for the Fourier transform on the infinite lattice, where $k \in [-\pi, \pi]$, in the main text. The obey the anticommutation relations $\{c(k), c^\dagger(k')\} = 2\pi\delta(k - k')$.

It is of course also possible to consider continuous real space fermions, in which case we use the notation $c^\dagger(x)$, which obeys the anticommutation relations $\{c(x), c^\dagger(x')\} = \delta(x - x')$. In the main text, $k$ and $q$ always refer to momentum while $x$ and $y$ always refer to position, so it is not possible to get them confused.

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
