# Peer review of "Extreme boundary conditions and random tilings"

_SciPost Physics Lecture Notes, doi:SciPost Phys. Lect. Notes 26 (2021)_

## Round 1 · Referee Report · Anonymous (Referee 1) · 2020-8-23

# Report on *Extreme boundary conditions and random tilings*

## 1   Overview

Random tiling (or dimer) model have a long history in Statistical Physics and probability theory, and to this day the object of intense interest. Thanks in particular to specific solvability properties (representation as free fermions or, in another language, determinantal point process), one can access a number of features (limit shapes, fluctuations in the bulk and near the boundary, ...) believed to hold for more general height models, such as the six-vertex model.

In these lecture notes, the author surveys results, heuristics and conjectures for tilings and related models, and explains how to derive some of them, mostly from the perspective of free fermions. More discussion of other approaches or formulations would certainly be useful; for example, much is known on dimers with Yang-Baxter weights, lattice realizations of fermions, use of algebraic combinatorics, non-intersecting path representations, etc. There is a fair amount of typos, some listed below, and the manuscript would also benefit from proof-reading for language. Otherwise, this is a generally clear, interesting, enjoyable read that exemplifies fundamental phenomena of current interest in Statistical Physics; in my opinion this is suitable for publication in SPLN after a revision.

## 2   Comments

- Page 7 Line 4: maybe emphasize which algorithms work only for specific boundary conditions.

- Eq. (11): the integration domain should exclude the diagonal

- below Eq. (17): "a solution": here and elsewhere discuss uniqueness of solutions of variational problems

- above Eq. (21): how does one rule out that the frozen region has e.g. 3 intervals ?

- P1L11: coulomb: Coulomb

- P13L8: "integers": discuss somewhere the different conventions on heights (RSOS or otherwise)

- above Figure 6: the color convention was discussed earlier.

- P14L12: not immediately clear what the figure means

- above Eq (26): maybe mention that there is no loss of generality in choosing $a, 1/a, b, 1/b$ as weights (gauge equivalence).

- Figure 7 caption and elsewhere: "hop around": wrap around

- Eq. (28): say something about the existence of the limit

- three lines above (32): "figure."

- P17L2: "zero mode": not sure this is an accurate terminology here (with boundary conditions)

- below (38) and throughout: "euclidean": Euclidean (similarly, Gaussian, Hermitian, etc.)

- footnote 4: GMC is not synonymous with GFF; $c = 1$ CFT is also dubious (especially in the dimer context, which is also closely related to $c = -2$ CFTs).

- paragraph below (39), last sentence: "essentially the same": what, if any, distinction is there ?

- P19L1: "BKT": expand acronyms on first occurrence.

- P18 first bullet point: the discussion here is unclear and perhaps dubious.

- Exercise 2.2.5: "odd case": what is the odd case ? The heuristics in this exercise are not terribly clear, in particular the position of $K$ in the last displayed equation

- Figure 10 caption: "occupation": occupancy

- below Figure 10: "thiner": spelling

- discussion around (58) is unclear

- below (65): explain what antiperiodic conditions are

- below (67): "integrant": sp.

- below (75) and elsewhere: "explicitely": sp.

- below (84): independent of $r$ and $s$: comment on whether this is a general or model-specific fact.

- below (90): "in section 4". this is section 4 ??

- below (92): "finding from": something missing

- two lines below (103): is there something missing in the LHS ?

- six lines below (106): "point": vertex ?

- P40L-7: resembles: the resemblance is rather superficial (e.g. different tails etc.)

- P43L4: "In turns out"

- P50L10: "nontriaval"

- below (150): explain the notation $(-1)^{(matrix)}$

- A.2 line 1: "droping": sp. ; "good physicists": this can also be couched in terms of distinguishing between an algebra and a representation of said algebra.

- below (159): define $\{,\}$

- References: some capitalization issues (ising, harnack, burgers etc), some missing journal information.

---

## Round 2 · Author Response

I am happy to resubmit a new version of the notes which addresses the points raised by the referee. Below is a detailed answer, together with the changes made to the manuscript.

---

## Round 2 · List of Changes

1) Page 7 Line 4: maybe emphasize which algorithms work only for specific boundary conditions.
Answer: done.
2) Eq. (11): the integration domain should exclude the diagonal
Answer: I added a comment stating that the aforementioned singularity is integrable.
3) below Eq. (17): “a solution”: here and elsewhere discuss uniqueness of solutions of variational problems
Answer: the point of this section is not really to discuss general minimization problems, rather to give an intuitive idea how to find the solution. That being said, I added a comment saying that uniqueness is ensured modulo some convexity condition.
4) above Eq. (21): how does one rule out that the frozen region has e.g. 3 intervals ?
Answer: the argument presented here does not formally rule that out. However, the particles tend to accumulate near the endpoints of the interval, so this does not seem physically plausible. I refer to Ref. [17] for a proof of this result.
5) P13L8: “integers”: discuss somewhere the different conventions on heights (RSOS or otherwise)
Answer: in various places in the text, I tried to clarify the conventions used.
6) above Figure 6: the color convention was discussed earlier
Answer: indeed, I clarified that.
7) P14L12: not immediately clear what the figure means
Answer: I tried to clarify that.
8) above Eq (26): maybe mention that there is no loss of generality in choosing a,1/a, b,1/b as weights (gauge equivalence).
Answer: this is true, but this is not really necessary to the argument I'm making.
9) Figure 7 caption and elsewhere: “hop around”: wrap around
Answer: this is fixed now.
10) three lines above (32): “figure.”
Answer: this is fixed now.
11) P17L2: “zero mode”: not sure this is an accurate terminology here (with boundary conditions)
Answer: I removed the term, which is indeed not needed.
12) below (38) and throughout: “euclidean”: Euclidean (similarly, Gaussian,Hermitian, etc.)
Answer: fixed, in various places in the text.
13) Footnote 4: GMC is not synonymous with GFF;c= 1 CFT is also dubious(especially in the dimer context, which is also closely related toc=−2CFTs).
Answer: I clarified that GMC is not synonymous with with GFF. The meaning of this footnote should not be taken too literally. The aim was simply to tell the reader that all those objects are very related, with various degrees of mathematical rigor, and different names depending on your field of research. For example the name GFF is perhaps the most standard in math, but it can be confusing for a physicist (free means gaussian in physics parlance).
The claim that the GFF or FF is a c=1 CFT is not at all dubious, it is a basic fact about CFT. See for example the following lecture notes by Cardy, 0807.3472, page 11, after (16). That being said, it is possible to tweak the free field to attain other central charges, as discussed in section 6.2 of the same lecture notes.
It is true that dimers can be related to c=-2 CFTs, if one wishes to do so. The way this is done is by constructing another transfer matrix, and checking that it is compatible with such CFTs. This other transfer matrix is different from the one introduced in section 3, which has the advantage of being Hermitian --which is not compatible with negative central charge-- and gives c=1. Similarly, the height mapping leads much more naturally to c=1.
Therefore, the present lecture notes fit much better with the c=1 point of view, which is arguably the most widely encountered CFT. It was a conscious decision on my part not do discuss those subtleties in these notes, even though I do agree some of the physics literature on the topic can be a little confusing.
14) paragraph below (39), last sentence: “essentially the same”: what, if any, distinction is there ?
Answer: there is no difference. I removed the word "essentially", which is perhaps confusing.
15) P19L1: “BKT”: expand acronyms on first occurrence.
Answer: fixed.
16) P18 first bullet point: the discussion here is unclear and perhaps dubious.
Answer: I am not certain which bullet point the referee refers to.
For the first bullet point on page 17, this is a typical non-rigorous physicist argument, which leads to unambiguous predictions (such as the critical exponent for dimer-dimer correlations is 2), which require considerably more work to prove.
For the only bullet point on page 18, this has been known for a long time in the physics literature, but I also give references to two recent math papers.
17) Exercise 2.2.5: “odd case”: what is the odd case ? The heuristics in this exercise are not terribly clear, in particular the position of K in the last displayed equation
Answer: the definition of what I mean by even and odd was missing, which made the exercise indeed not very clear.
18) Figure 10 caption: “occupation”: occupancy
Answer: this is fixed now.
19) below Figure 10: “thiner”: spelling
Answer: this is fixed now.
20) discussion around (58) is unclear
Answer: I agree. I significantly rewrote and expanded the corresponding paragraph.
21) below (65): explain what antiperiodic conditions are
Answer: I agree this was not very clear. I chose to remove the reference to antiperiodic boundary conditions, since those are not really needed.
22) below (67): “integrant”: sp.
Answer: this is fixed now.
23) below (75) and elsewhere: “explicitely”: sp.
Answer: this is fixed now.
24) below (84): independent ofrands: comment on whether this is a general or model-specific fact.
Answer: this holds only for model which can be mapped to free fermions. This identity does not hold for more generic models, such as interacting dimers. I added a sentence to make that point clearer.
25) below (90): “in section 4”. this is section 4 ??
Answer: I meant section 4.5. This is fixed now.
26) below (92): “finding from”: something missing
Answer: I meant finding it from. This is fixed now.
27) two lines below (103): is there something missing in the LHS ?
Answer: this is fixed now.
28) six lines below (106): “point”: vertex ?
Answer: I'm not sure what the referee refers to.
29) P40L-7: resembles: the resemblance is rather superficial (e.g. different tails etc.)
Answer: indeed this is superficial. I added a comment to that effect.
30) P43L4: “In turns out”
Answer: this is fixed now.
31) P50L10: “nontriaval”
Answer: this is fixed now.
32) below (150): explain the notation (-1)^(matrix)
Answer: this is meant in the sense of matrix exponential.
33) A.2 line 1: “droping”: sp. ; “good physicists”: this can also be couched in terms of distinguishing between an algebra and a representation of said algebra.
Answer: true. This comment was perhaps needlessly provocative, I decided to remove it.
34) below (159): define{,}
Answer: this is fixed now.
35) References: some capitalization issues (ising, harnack, burgers etc), somemissing journal information.
Answer: this is fixed now.
In addition to that, I made a few other changes, the most significant of which are listed below.
a) I added two long exercises treating dimers on the square lattice. While the introduction was centered on square dimers, the later parts of the notes only treated dimers on honeycomb, which are slightly simpler. The reader wanting to apply the method of sections 3,4 to derive the results exposed in the introduction can do so by solving those two exercises.
b) I corrected some typos in section 3, including one in the action of the transfer matrix, which had a transpose gotten wrong.

---

## Editorial Decision

published